# QUIRKY regulates root epidermal cell patterning through stabilizing SCRAMBLED to control CAPRICE movement in *Arabidopsis*

Jae Hyo Song [1], Su-Hwan Kwak[2], Kyoung Hee Nam[3], John Schiefelbein [4] & Myeong Min Lee [1]

SCM, a leucine-rich repeat receptor-like kinase, is required for root epidermal cells to appropriately interpret their location and generate the proper cell-type pattern during *Arabidopsis* root development. Here, via a screen for *scm*-like mutants we describe a new allele of the *QKY* gene. We find that QKY is required for the appropriate spatial expression of several epidermal cell fate regulators in a similar manner as SCM in roots, and that QKY and SCM are necessary for the efficient movement of CPC between epidermal cells. We also show that turnover of SCM is mediated by a vacuolar degradation pathway triggered by ubiquitination, and that QKY prevents this SCM ubiquitination through their physical interaction. These results suggest that QKY stabilizes SCM through interaction, and this complex facilitates CPC movement between the epidermal cells to help establish the cell-type pattern in the *Arabidopsis* root epidermis.

[1] Department of Systems Biology, Yonsei University, 50 Yonsei-ro, Seoul 03722, Korea. [2] Department of Biology, Long Island University, 1 University Plaza, Brooklyn, NY 11201, USA. [3] Department of Biological Science, Sookmyung Women's University, Cheongpa-ro 47-gil 100, Yongsan-gu, Seoul 04310, Korea. [4] Department of Molecular, Cellular, and Developmental Biology, University of Michigan, Ann Arbor, MI 48109, USA. Correspondence and requests for materials should be addressed to M.M.L. (email: mmlee@yonsei.ac.kr)

The specification of distinct cell fates is a critical process in the development of multicellular organisms. In many cases, cell fate decisions are influenced by the relative position of a cell to its neighbors, indicating that cell–cell communication is crucial[1–3]. A simple model system for the study of cell fate specification is found in the *Arabidopsis* root epidermis, which is composed of two cell types, root hair-bearing cells (hair cells) and non-hair cells, that are patterned in a position-dependent manner[4,5]. The epidermal cells located outside a periclinal cortical cell wall (N position) contacting a single cortical cell differentiate into non-hair cells, while the epidermal cells located over an anticlinal cortical cell wall (H position) contacting two underlying cortical cells differentiate into hair cells.

Many genes are known to influence cell fate specification in the *Arabidopsis* root epidermis. *GLABRA2* (*GL2*), which encodes a homeodomain-leucine zipper (HD-Zip) protein, is expressed in developing epidermal cells in the N position and specifies the non-hair cell fate[6]. *WEREWOLF* (*WER*) and *CAPRICE* (*CPC*) encode an R2R3 MYB-type transcription factor and an R3 single MYB-type transcription factor, respectively, and regulate *GL2* expression competitively in a dose-dependent manner[7–9]. *WER* is expressed preferentially in the developing N-position cells and directly induces *GL2* expression to specify the non-hair cell fate, whereas CPC inhibits *GL2* expression in the H-position cells to specify the hair cell fate. Interestingly, WER is a direct positive regulator of *CPC* in the N-position cells[10], and CPC protein moves to the neighboring H-position cells[11] to repress the expression of *WER* and *GL2*[12]. Further, the H-cell accumulation of CPC is regulated, in part, by the ENHANCER OF GLABRA3 (EGL3), a basic helix-loop-helix protein[13]. Thus, the differentiating epidermal cells communicate with each other to achieve their proper cell-type pattern via the action of a mobile transcription factor (CPC) produced by WER-expressing N-cells to prevent neighboring H-cells from adopting the non-hair cell fate. The intercellular movement of CPC is likely to occur through plasmodesmata (PD), which provide cytoplasmic connections between plant cells[14,15].

In addition to intercellular trafficking, plant cells communicate with other cells and their environment through receptor-like kinases (RLKs) on the cell surface. A leucine-rich repeat receptor-like kinase (LRR RLK), SCRAMBLED (SCM; also known as STRUBBELIG), is required for proper cell-type patterning in the

*Arabidopsis* root epidermis[16], as well as outer integument development in the ovule[17], fruit dehiscence[18], internode growth[17], and tissue morphogenesis[17,19]. In the developing root epidermis, SCM accumulates preferentially in the H-position cells through a feedback mechanism[20], and has been proposed to respond to a positional signal and preferentially inhibit *WER* expression in the H-position cells[21]. However, it is not yet known how the initial difference in SCM activity between the N-position cell and the H-position cell is initiated. Furthermore, it is not clear how SCM action leads to inhibition of *WER* expression in the H-position cell, considering that SCM kinase activity is not required for epidermal cell patterning[17,18].

To understand how SCM functions in root epidermal cell patterning, we used a genetic approach to search for new regulators acting in the SCM signaling pathway. We identified a mutant with an *scm-like* root mutant phenotype, and found that it is an allele of the *QUIRKY* (*QKY*) gene. QKY, a multiple C2 domain and transmembrane region protein (MCTP), has previously been reported to interact with SCM and affect epidermal cell patterning[22–24], although its role in root epidermal cell patterning is unknown. Here, we show that QKY acts upstream of WER and GL2, and we find that QKY regulates the efficient movement of CPC between epidermal cells through stabilizing SCM protein at the plasma membrane (PM). Furthermore, we demonstrate that QKY and SCM physically interact through their extracellular domain, and that QKY prevents the ubiquitination of SCM which triggers SCM degradation in vacuoles.

## Results

**Isolation of a new allele of *qky*.** We screened a population of ethylmethanesulfonate (EMS) mutagenized seeds harboring the *GL2p:GFP* marker and isolated a mutant showing defects in position-dependent root epidermal patterning and expression of the *GL2p:GFP* marker (Supplementary Fig. 1a, b and Table 1). We confirmed that this phenotype is caused by a single nuclear recessive mutation by analyzing the F1 and F2 offspring from a cross with wild-type plants. Through a bulk segregant analysis, we found that the mutation is linked to a marker (nga111) on chromosome 1, which is near the *QKY* gene previously reported to affect root epidermal cell patterning[22]. Allelism testing (by crossing this new mutant with *qky-8*[22] and with *scm-2*[16]) showed

**Table 1 Specification of cell types in the root epidermis**

| Genotype | H at H pos (%) | N at H pos (%) | H at N pos (%) | N at N pos (%) |
|---|---|---|---|---|
| Wild type | 98.0 ± 1.0 | 2.0 ± 1.0 | 0.0 ± 0.0 | 100.0 ± 0.0 |
| *qky-16* | 74.5 ± 4.0 | 25.5 ± 4.0 | 18.2 ± 3.2 | 81.8 ± 3.2 |
| *qky-16 QKYp:QKY* | 95.3 ± 0.6 | 4.7 ± 0.6 | 4.3 ± 1.2 | 95.7 ± 1.2 |
| *qky-16 QKYp:QKY-GFP* | 98.0 ± 1.0 | 2.0 ± 1.0 | 0.3 ± 0.6 | 99.7 ± 0.6 |
| *qky-16 CO2p:QKY-GFP* | 71.5 ± 5.5 | 28.5 ± 5.5 | 21.4 ± 6.2 | 78.6 ± 6.2 |
| *qky-16 SHRp:QKY-GFP* | 69.4 ± 4.5 | 30.6 ± 4.5 | 22.5 ± 6.4 | 77.5 ± 6.4 |
| *qky-16 SCRp:QKY-GFP* | 72.3 ± 4.0 | 27.7 ± 4.0 | 28.2 ± 5.8 | 71.8 ± 5.8 |
| *qky-16 WERp:QKY-GFP* | 96.0 ± 1.5 | 4.0 ± 1.5 | 0.0 ± 0.0 | 100.0 ± 0.0 |
| *scm-2* | 69.0 ± 3.6 | 31.0 ± 3.6 | 21.7 ± 2.2 | 78.3 ± 2.2 |
| *qky-16 scm-2* | 66.6 ± 6.2 | 33.4 ± 6.2 | 24.5 ± 5.3 | 75.5 ± 5.3 |
| *RHD3p:QKY* | 97.0 ± 1.5 | 3.0 ± 1.5 | 0.0 ± 0.0 | 100.0 ± 0.0 |
| *RHD3p:QKY qky-16* | 95.0 ± 1.2 | 5.0 ± 1.2 | 2.0 ± 1.0 | 98.0 ± 1.0 |
| *RHD3p:QKY scm-2* | 71.5 ± 6.2 | 28.5 ± 6.2 | 22.6 ± 3.0 | 77.4 ± 3.0 |
| *cpc-1* | 30.3 ± 1.5 | 69.7 ± 1.5 | 0.7 ± 0.6 | 99.3 ± 0.6 |
| *qky-16 cpc-1* | 20.7 ± 1.2 | 79.3 ± 1.2 | 2.7 ± 1.2 | 97.3 ± 1.2 |
| *SCMp:SCM$_{Em1}$-GFP scm-2* | 91.7 ± 2.5 | 8.3 ± 2.5 | 3.3 ± 0.6 | 96.7 ± 0.6 |
| *SCMp:SCM$_{Em2}$-GFP scm-2* | 76.0 ± 2.7 | 24.0 ± 2.7 | 17.3 ± 4.0 | 82.7 ± 4.0 |

Values represent means ± standard deviations (SD) of three independent experiments. In each experiment, ten 4-day-old seedlings were examined for each strain except for the *qky-16* mutant (eleven 4-day-old seedlings were examined for this strain)
*H at H pos* hair cells at H position, *N at H pos* non-hair cells at H position *H at N pos* hair cells at N position, *N at N pos* non-hair cells at N position

that the mutant phenotype was complemented by *scm-2* but not complemented by *qky-8* (Supplementary Fig. 1c). We sequenced the *QKY* coding region in the genomic DNA from this mutant, which revealed a nonsense mutation at the 870th codon (Supplementary Fig. 1d). In addition, we discovered that a *QKY* genomic DNA fragment including 1.2 kb 5′- and 1 kb 3′-flanking sequences (*QKYp:QKY*) completely rescued the abnormal root epidermis phenotype of the new mutant (Supplementary Fig. 1e and Table 1). Together, these results indicate that this newly identified mutant is a new allele of the *QKY* gene, and we named it *qky-16*.

**Expression of cell fate regulators in the *qky* mutant root**. To determine the regulatory relationship between QKY and previously identified transcriptional regulators of the root epidermis pathway, we examined the promoter activity of *WER*, *CPC* and *EGL3* using transcriptional reporter genes (*WERp:GFP*, *CPCp:GUS*, and *EGL3p:GUS*) in the *qky-16* mutant. In the wild-type root, the *WERp:GFP* and the *CPCp:GUS* are preferentially expressed in the N-position cells, while *EGL3p:GUS* is preferentially expressed in the H-position epidermal cells[7,25,26]. In the *qky-16* mutant, the position-dependent expression pattern of these three genes was disrupted, causing reporter gene-expressing cells and reporter gene-non-expressing cells to be produced at each position (Supplementary Fig. 1a), which is similar to the expression patterns in the *scm-2* mutant root[16]. These results suggest that, like SCM, QKY acts early during root epidermis development to enable cells to interpret their relative position and appropriately regulate expression of the downstream transcription factor network.

**QKY expression pattern in the root epidermis**. To evaluate *QKY* gene expression, we generated transgenic plants bearing a *QKYp:GUS* transcriptional reporter construct (*GUS* gene fused to the same *QKY* regulatory sequences used for the genomic DNA complementation described above). These plants exhibit strong GUS accumulation in the stele cells and the ground tissue in the root meristematic region, as well as a lower GUS level in developing root epidermal cells and root cap cells (Fig. 1a), consistent with previous reports of widespread expression of *QKY* in all organs[23,24].

The QKY protein has been reported to accumulate in the PM in most of the tissues in the root using an N-terminal eGFP fusion to QKY (eGFP-QKY)[23]. We generated translational fusions of GFP to the N-terminus or C-terminus of QKY and expressed these under the control of the *QKY* regulatory sequences (the same 5′ and 3′ regions used in the *QKYp:GUS* construct), but these constructs failed to complement the *qky-16* mutant root phenotype (Supplementary Fig. 1e). Therefore, we tested several versions of reporter constructs containing *GFP* in-frame with the *QKY* coding sequence, and found that a translational reporter construct with the *GFP* sequence inserted between the end of the first transmembrane domain and the beginning of the phosphoribosyltransferase (PRT) domain (*QKYp:QKY-GFP*) was able to completely rescue the *qky-16* mutant phenotype (Supplementary Fig. 1e and Table 1). As expected, the fluorescence signal in this line was detected at the PM of the cells in most of the root tissues including the epidermis, which is similar to the SCM (*SCMp:SCM-GFP*) localization pattern (Fig. 1b)[20]. The QKY-GFP signal was unevenly distributed in the PM, which is consistent with the previously reported PD-localization of QKY[23]. Interestingly, we discovered greater accumulation of QKY-GFP in the H-position cells than the N-position cells of the root epidermis (Fig. 1b),

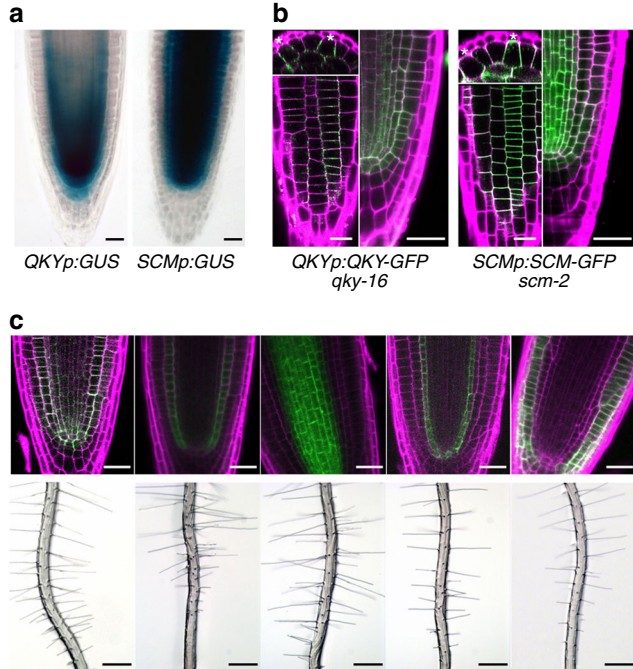

**Fig. 1** Expression pattern and tissue-autonomous function of *QKY* gene in the root. **a** The promoter activity of *QKY* (*QKYp:GUS*) and *SCM* (*SCMp:GUS*) in the developing root of 4-day-old seedlings. Bar = 25 μm. **b** Localization of QKY and SCM in the developing root meristem. Four-day-old seedlings bearing *QKYp:QKY-GFP* and *SCMp:SCM-GFP* translational fusions (left top, transverse section; left bottom, surface view; right, median longitudinal section) were stained with propidium iodide (magenta) and analyzed for GFP accumulation (green). The photos for each view were taken at a similar location in the root, respectively. Asterisks indicate the H-position cells. Bar = 25 μm. **c** Tissue-specific expression of QKY-GFP in *qky-16* mutant. Four-day-old seedlings of *qky-16* plants harboring different complementation constructs were stained with propidium iodide (magenta) and analyzed for QKY-GFP accumulation (green) (top; bar = 25 μm). Root hair phenotype of *qky-16* seedlings harboring different complementation constructs (bottom; bar = 250 μm)

Figure panel labels: QKYp:GUS | SCMp:GUS | QKYp:QKY-GFP qky-16 | SCMp:SCM-GFP scm-2 | QKYp:QKY-GFP qky-16 | CO2p:QKY-GFP qky-16 | SHRp:QKY-GFP qky-16 | SCRp:QKY-GFP qky-16 | WERp:QKY-GFP qky-16

implying a preferential role for QKY function in the differentiating hair cells during root epidermis patterning.

Due to the broad expression of *QKY* in root tissues and a report suggesting non-cell autonomous action of QKY in aerial tissues[23], we tested the long-range action of QKY in regulating root epidermal cell patterning. We expressed QKY-GFP only in the stele, endodermis, cortex, or epidermis/lateral root cap tissues of the *qky-16* mutant by using the *SHORTROOT* (*SHR*)[27], *SCARECROW* (*SCR*)[28], *CO2*[29], or *WER*[7] promoter, respectively. We observed the fluorescence signal only in the tissue(s) where each promoter was expected to be active, and we found that only the *WERp:QKY-GFP* construct was able to rescue the *qky-16* mutant phenotype in the root epidermis (Fig. 1c and Table 1). These results indicate that QKY influences root epidermal cell patterning in a tissue autonomous manner. Notably, SCM has been reported to act in a cell autonomous manner in the root epidermis[20].

**Genetic interaction between QKY and SCM**. Given the similar mutant phenotypes, gene expression patterns, protein accumulation, and epidermis action for *QKY* and *SCM*, we sought to examine the relationship between these two genes. First, we tested for possible genetic interaction by generating the *qky-16 scm-2*

**a**

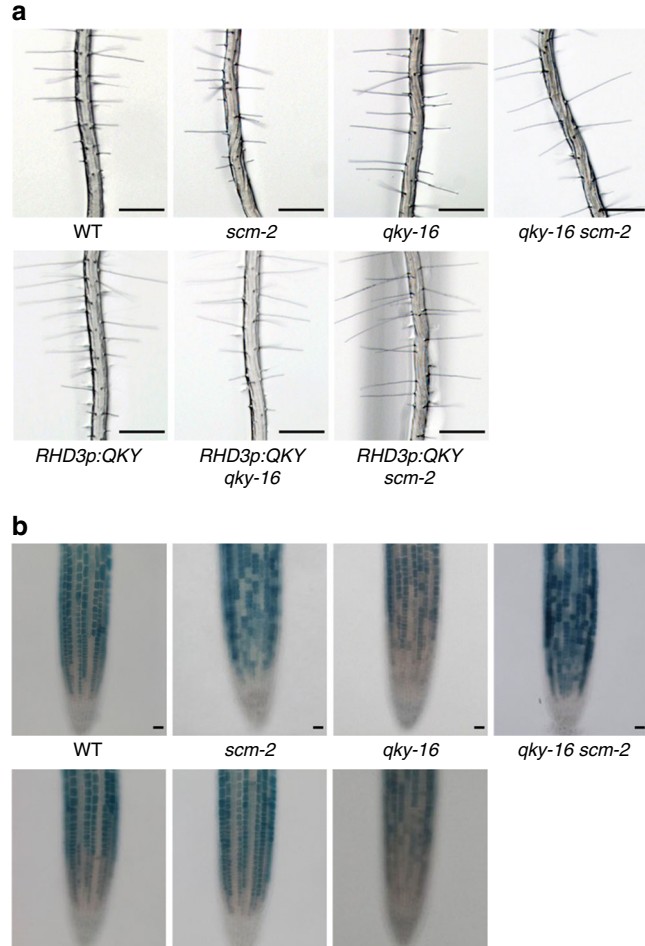

**b**

**Fig. 2** QKY and SCM act in the same genetic pathway. **a** Root hair phenotype in the root of various mutant plants. Bar = 250 μm. **b** GL2p:GUS expression pattern in the root of the corresponding various mutant plants. Bar = 25 μm

double mutant by a genetic cross. The abnormal epidermal cell pattern in this double mutant was not significantly different from the abnormal pattern in each of the single mutants (Fig. 2a and Table 1). Similarly, the double mutant exhibited disruption in the position-dependent expression of the *GL2p:GUS* marker that was comparable to the single mutants (Fig. 2b). *QKY* overexpression using the promoter of the *ROOT HAIR DEFECTIVE 3* (*RHD3*) gene, which shows strong activity ubiquitously in the root (*RHD3p:QKY*)[30], was able to rescue the *qky-16* mutant root phenotype completely, but it was not able to rescue the *scm-2* mutant phenotype and did not affect position-dependent cell-type patterning in the wild-type root epidermis (Fig. 2 and Table 1). Together, these results suggest that *QKY* and *SCM* act in a same genetic pathway to regulate root epidermal cell patterning, and that *QKY* functions upstream of *SCM* or they function in parallel.

**QKY and SCM regulate CPC movement between epidermal cells.** The disrupted cell patterning in the *qky* and *scm* mutants and the presumed localization of the QKY and SCM proteins at PD[23] led us to examine their possible role in the lateral inhibition between epidermal cells mediated by the mobile transcription factor CPC[12]. First, we examined the root epidermis phenotype of

the *qky-16 cpc-1* double mutant, and found that it was largely similar to the phenotype of *cpc-1* and quite different from the phenotype of the *qky-16* mutant (Table 1), implying that the *qky-16* mutant phenotype seems to be mediated by CPC at least in part. It was also shown that the *scm-2 cpc-1* root phenotype was quite different from the *scm-2* mutant phenotype[21].

Next, we tested whether QKY or SCM affects CPC movement using the *SHRp:CPC-GFP* line, in which stele-produced CPC-GFP fusion protein is able to move to the epidermis and preferentially accumulate in the H-position cells[13]. In both the *qky-16* and the *scm-2* mutant roots, the CPC-GFP protein produced by the *SHRp:CPC-GFP* construct was detected in the epidermal cells (Fig. 3a), indicating that QKY and SCM are not essential for radial trafficking of CPC. However, preferential accumulation of CPC-GFP in the H-position cells was disrupted in these mutants (about 40% of CPC-GFP accumulating epidermal cells were located at the N-position while about 10% in the wild-type root) (Fig. 3b), suggesting that QKY and SCM are required for preferential CPC trafficking or accumulation into the H-position cells.

We used a biolistic particle delivery system to produce GFP or the CPC-GFP fusion protein in a single epidermal cell and monitor its movement. When GFP alone was delivered and expressed in an epidermal cell of wild-type leaves, the GFP signal was also detected in a neighboring cell in more than 70% of the cases after 4 h (Fig. 3c and Supplementary Fig. 2a). Similar mobility of GFP was observed in the *qky-16* or *scm-2* mutants, consistent with a previous report, suggesting no general effect of QKY and SCM on GFP movement between cells[23]. The CPC-GFP fusion protein (*35Sp:CPC-GFP*) was also detected in neighboring cells in about 70% of the cases in wild-type leaves, but only in 46 and 38% of the cases in the *qky-16* and *scm-2* mutant leaves, respectively (Fig. 3c), implying that QKY and SCM affect CPC-GFP movement between epidermal cells. To determine whether SCM and QKY are involved in export of CPC-GFP from the *CPC-GFP* expressing cell, we coexpressed *SCM-CFP* or *QKY-CFP* together with *CPC-GFP* in the same leaf epidermal cell (in the *scm-2* or *qky-16* mutant, respectively) and analyzed CPC-GFP mobility. We found that CPC-GFP protein movement was not enhanced by coexpression of *SCM* or *QKY* (Fig. 3c and Supplementary Fig. 2b), implying that SCM and QKY do not affect export of CPC, so they rather may be involved in the import of CPC. This possibility is consistent with the greater abundance of SCM and QKY in the H-position cells of the root epidermis which preferentially accumulate the mobile CPC protein (Fig. 1b)[20]. It appears unlikely that SCM and QKY affect CPC movement through direct interaction, because we found that neither the cytoplasmic domain of SCM ($SCM_{CD}$) nor the three domains of QKY divided by the two transmembrane domains ($QKY_{D1}$, $QKY_{D2}$, and $QKY_{D3}$) are able to interact with CPC in yeast cells (Fig. 4a and Supplementary Fig. 3).

Given that CPC mobility is also important for proper trichome spacing in the shoot epidermis[31], we examined the leaf surface of the *qky, scm*, and *qky scm* mutants and discovered instances of adjacent "twin" trichomes (indicating defective CPC-mediated lateral inhibition) in the *qky-16* (0.54%, *n* = 925), the *scm-2* (0.55%, *n* = 900), and the *qky-16 scm-2* mutant (0.70%, *n* = 849), which was never observed in the wild-type leaves (*n* = 1376) (Supplementary Fig. 1f). This result further supports the view that QKY and SCM influence CPC movement between epidermal cells.

**QKY and SCM interact through their extracellular domains.** It has been reported that QKY and SCM physically interact with each other using the FRET-FLIM method and the yeast

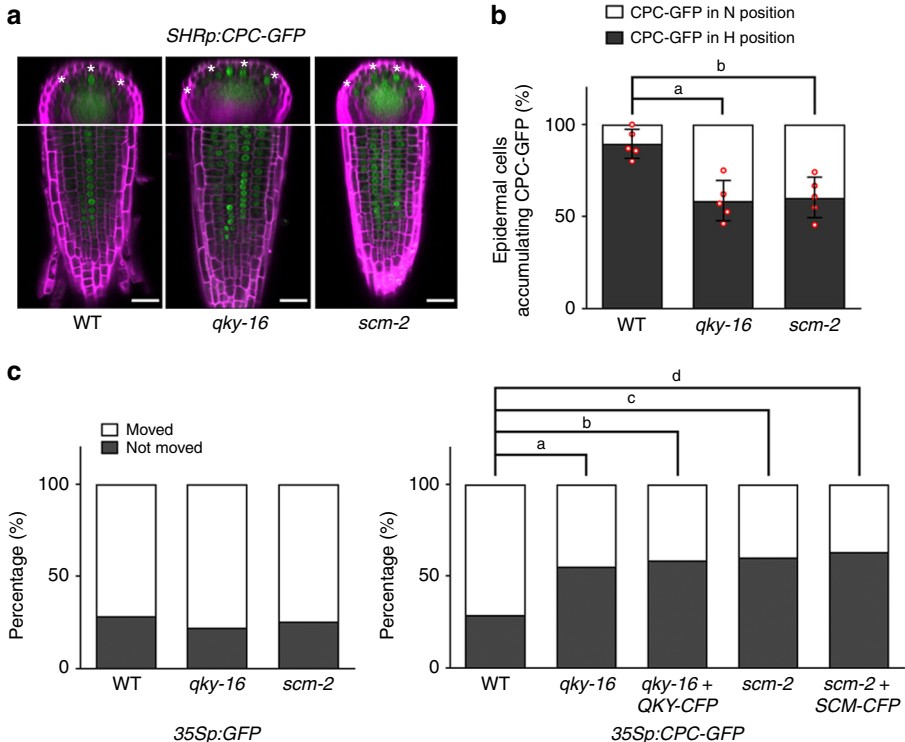

**Fig. 3** QKY and SCM control CPC movement between epidermal cells. **a** CPC-GFP accumulation in the root epidermis of various plants harboring the *SHRp: CPC-GFP* translational fusion. The H-position epidermal cells are marked with an asterisk. Bar = 25 μm. **b** Quantitative analysis of the stele-produced CPC-GFP movement to the epidermis of (**a**). Values represent means ± SD of five individual 4-day-old seedlings. In each seedling, at least 20 cells accumulating CPC-GFP were examined and their position was scored. Statistical significance was determined by one-way ANOVA with Bonferroni's post hoc test, and the *P* values for a and b are 0.000756586988099 and 0.001224674739152, respectively. **c** Quantitative analysis of the CPC-GFP movement. GFP or CPC-GFP was transiently expressed in rosette leaves of indicated mutants using the Biolistic PDS-1000/He system and the leaves were incubated for their expression for 4 h. Quantitative analysis was performed using only sink cells in rosette leaves. Values indicate the percentage of cells showing GFP movement to the neighboring cells (white) and cells not showing GFP movement (black). At least 76 cells were counted for each experiment. Actual number of cells analyzed in each experiment can be found in Supplementary Fig. 2. Statistical significance of differences in the frequency was tested by chi-square test, and the *P* values for a, b, c, and d are 0.0000160, 0.0000490, 0.0000004, and 0.0000032, respectively

two-hybrid assay[23,24]. We further confirmed this interaction using coimmunoprecipitation of epitope tagged full-length proteins (QKY-FLAG and SCM-MYC) expressed in tobacco leaves (Fig. 4b). In addition, we used the yeast two-hybrid assay to determine that the SCM extracellular domain interacts with the N-terminal domain 1 of QKY (QKY$_{D1}$) (Fig. 4a–c) rather than the SCM cytoplasmic domain which was previously reported to be a QKY-interacting domain[23]. Further, we found that a mutated version of the SCM extracellular domain (at a proline-repeat region; SCM$_{Em2}$) was not able to interact with the QKY$_{D1}$ (Fig. 4a–d). We also found that this mutated version of SCM (*SCMp:SCM$_{Em2}$-GFP*) was not able to complement the *scm-2* mutant phenotype (Fig. 4e and Table 1). Interestingly, a mutation in a different proline-repeat region in the extracellular domain (SCM$_{Em1}$; *SCMp:SCM$_{Em1}$-GFP*) did not affect the functionality of SCM nor its interaction with QKY (Fig. 4a–e and Table 1).

These results suggest that the QKY$_{D1}$ could be an extracellular domain that interacts with the SCM extracellular domain. However, it was reported that the cytoplasmic domain of SCM interacts with the QKY$_{D1}$, leading to the suggestion that this is a cytoplasmic domain[23]. To examine the topology of QKY at the PM, we isolated protoplasts from *QKYp:QKY-GFP qky-16* or *SCMp:SCM-GFP scm-2* plants, applied proteinase K to digest the exposed extracellular domain of the membrane-spanning proteins, and performed western blot analysis with total protein extracts using anti-GFP antibodies (Fig. 4f). The proteinase K-

treated cells yielded a low level of full-length proteins (130 kD for SCM-GFP and 170 kD for QKY-GFP) and a large amount of truncated proteins (70 kD for SCM-GFP and 40 kD for QKY-GFP) in the western blot analysis, implying that the kinase domain of SCM (including the transmembrane domain) and the domain 2 of QKY (including the two transmembrane domains) fused to GFP were protected. We further evaluated the topology of QKY using the bimolecular fluorescence complementation (BiFC) assay (Fig. 4g). We fused the N-terminal half of EYFP (EYFPn) to the C-terminus (cytoplasmic domain) of SCM (SCM-EYFPn-V2), and we fused the C-terminal half of EYFP (EYFPc) between the end of the first transmembrane domain and the beginning of the following PRT domain in QKY-GFP (QKY-EYFPc-V2). We coexpressed these in *Nicotiana benthamiana* leaf epidermal cells and were able to detect EYFP fluorescence. On the contrary, the SCM-EFYPn-V2 was not able to show the fluorescence complementation with the QKY-EYFPc-V1 or QKY-EYFPc-V3 which has the EYFPc at the end of the N-terminus or the C-terminus of QKY, respectively (Fig. 4g). In a reciprocal experiment, we found that SCM-EYFPn-V1 (EYFPn fused to the N-terminus of SCM) did not show fluorescence complementation with QKY-EYFPc-V2 but did with QKY-EYFPc-V1 or QKY-EYFPc-V3 (Supplementary Fig. 4).

Together, these results suggest that the domain 1 and domain 3 of QKY are extracellular domains, and that domain 1 is used by QKY to interact with SCM and regulate epidermal cell patterning.

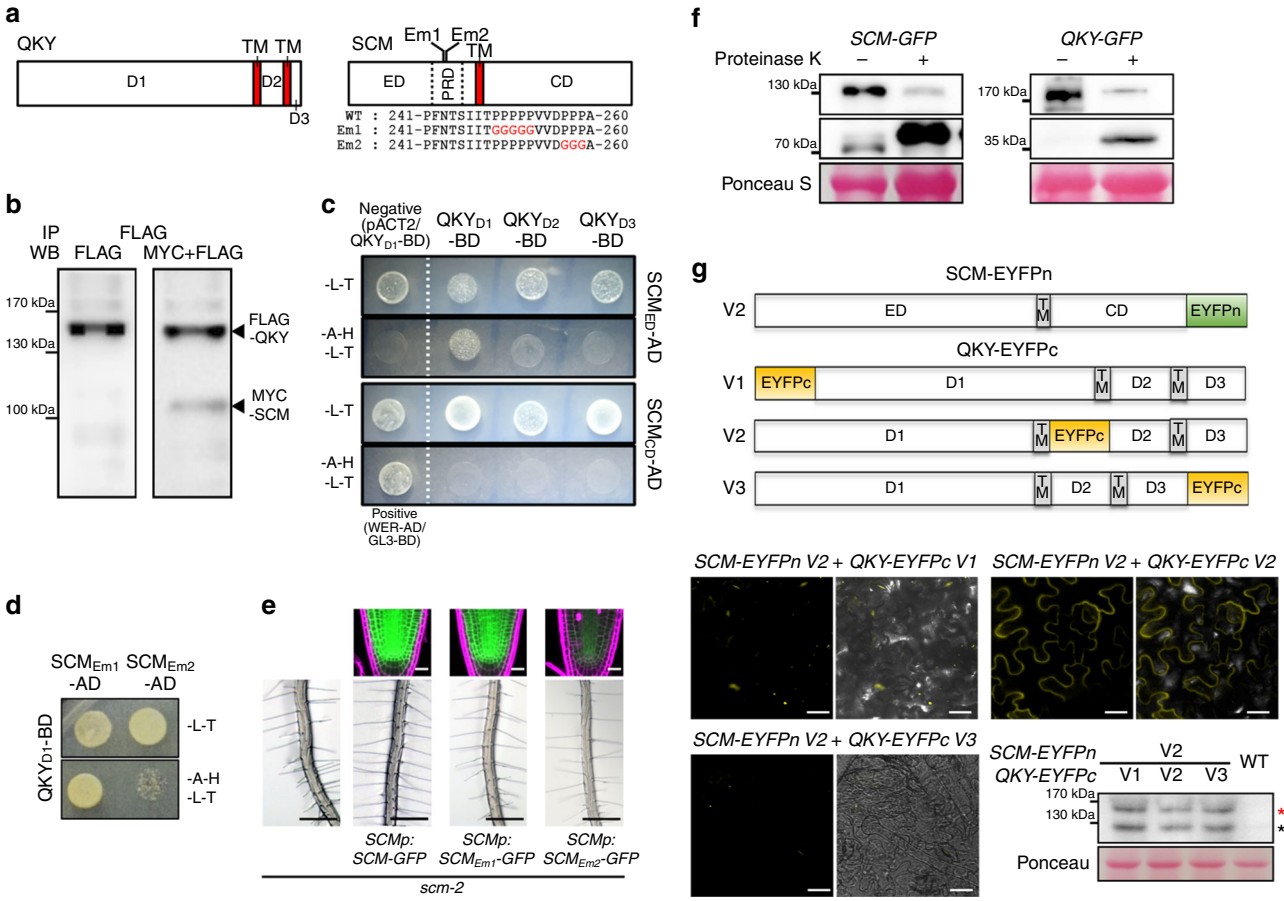

**Fig. 4** Interaction between QKY and SCM proteins. **a** Structure of the QKY and the SCM proteins. Schematic representation of QKY (left) shows the N-terminus domain (D1) and the other two domains (D2 and D3) separated by two transmembrane domains (TM). Schematic representation of SCM (right) shows the extracellular domain (ED) containing a proline-rich domain (PRD), and the cytosolic domain (CD), which are separated by one transmembrane domain (TM). **b** Coimmunoprecipitation of QKY with SCM protein. The experiment was performed with total protein extracts from *Nicotiana benthamiana* leaves transiently transformed with SCM-MYC and QKY-FLAG. **c** Yeast two-hybrid assay with QKY and SCM. The construct combinations were cointroduced into the yeast strain AH109. Transformants were grown on the -L-T (lacking leucine and tryptophan) control plates and the -A-H-L-T (lacking adenosine, histidine, leucine and tryptophan,) selective plates for 3 days. The combination of WER/GL3 was used as a positive control, and pACT2(empty vector)/QKY-D1 as a negative control (BD: pAS2-1; AD: pACT2). **d** Effect of mutations in the SCM PRD on the interaction between QKY and SCM in a yeast two-hybrid assay. Mutations are shown in the right panel of (**a**). **e** Effect of mutations in the SCM PRD on the SCM-GFP protein level (top, bar = 20 μm) and the root hair phenotype (bottom, bar = 250 μm). **f** Proteinase K protection test. Protoplasts from *SCMp:SCM-GFP* and *QKYp:QKY-GFP* stable transgenic plants were treated with proteinase K (1 ng/μl). Total protein extracts were subjected to immunoblotting using monoclonal anti-GFP antibodies. **g** BiFC assay with QKY and SCM. SCM-EYFPn-V2 in which the N-terminal half of EYFP (EYFPn) was fused to the C-terminus of SCM and QKY-EYFPc-Vx in which the C-terminal half of EYFP (EYFPc) was fused to various sites as shown in the upper panel were coexpressed in the *Nicotiana benthamiana* leaf epidermal cell using a biolistic particle delivery system. Their expression was confirmed by western blot analysis with total protein extracts using polyclonal anti-GFP antibodies. The red and the black asterisks indicate the QKY-EYFPc protein and the SCM-EYFPn protein, respectively. Bar = 25 μm

**QKY regulates the internalization of SCM protein**. To investigate the biological impact of QKY−SCM interaction, we examined SCM-GFP protein accumulation in the *SCMp:SCM-GFP qky-16 scm-2* plant root. We discovered that the level of SCM-GFP protein was much lower than in the *SCMp:SCM-GFP scm-2* plant root (Fig. 5a, b), while *SCM* promoter activity and *SCM-GFP* transcript level were not affected by the *qky* mutation (Supplementary Fig. 5). On the contrary, the *scm* mutation did not significantly affect QKY-GFP protein accumulation and *QKY-GFP* transcript level (Fig. 5a and Supplementary Fig. 5). In addition, the fluorescence intensity from the SCM_Em2-GFP (which fails to interact with QKY) was much weaker than the intensity from SCM-GFP and from SCM_Em1-GFP even though their transcript levels are similar (Fig. 4e and Supplementary Fig. 6). These results suggest that interaction between QKY and SCM is important for the stabilization of SCM protein.

Therefore, we examined the effect of MG132 (an inhibitor of proteasome and cysteine protease), lactacystin (a proteasome-specific inhibitor), and E-64d (a cysteine protease inhibitor) on SCM-GFP protein level. Treatment of MG132 and E-64d increased the SCM-GFP protein level in *qky-16 scm-2* and in *scm-2* mutant roots, while lactacystin did not affect SCM-GFP accumulation (Fig. 5c), indicating that SCM is rapidly degraded in a cysteine protease-dependent manner rather than by a proteasome in the *qky-16 scm-2* mutant. Interestingly, the major protein reacting with the anti-GFP antibodies in the *qky-16 scm-2* mutant root had higher molecular weight (MW) (red asterisk) than the SCM-GFP protein in the *scm-2* mutant root (black asterisk) (Fig. 5b). Upon treatment with MG132 and E-64d, the high MW SCM-GFP band (red asterisk) showed greater accumulation in the *qky-16 scm-2* mutant root compared with the same sized SCM-GFP in the *scm-2* mutant root (Fig. 5c).

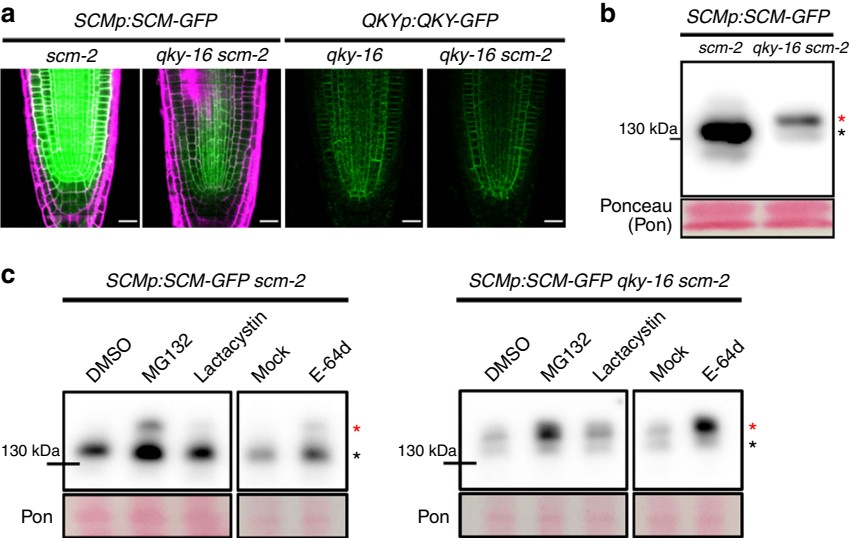

**Fig. 5** QKY regulates the level of SCM protein in the root. **a** GFP accumulation in the developing root meristem of various plants harboring *SCMp:SCM-GFP* (left) and *QKYp:QKY-GFP* (right) translational fusions. Bar = 20 µm. **b** Western blot analysis to examine the SCM-GFP level using monoclonal anti-GFP antibodies. **c** Western blot analysis to examine the SCM-GFP level. Four-day-old seedlings were incubated with MG132 (50 µM), lactacystin (50 µM) or E-64d (20 µM) for 2 h. Total protein extracts were subjected to immunoblotting using monoclonal anti-GFP antibodies

SCM-GFP preferentially accumulated in the PM, and E-64d treatment increased internal accumulation of SCM-GFP in subcellular membranous compartments much more in the *qky-16 scm-2* mutant root than in the *scm-2* mutant root (Fig. 6a). Because E-64d is known to inhibit the fusion of late endosomes with vacuoles in *Arabidopsis*[32], we examined whether SCM-GFP accumulation in late endosomes increased after E-64d treatment. We introduced the *UBQ10p:mCherry-RabF2a*, a marker of late endosomes[33], into the *SCMp:SCM-GFP qky-16 scm-2* plant by a genetic cross, and discovered that a small portion of the internal SCM-GFP colocalized with mCherry-RabF2a without E-64d treatment (DMSO) (Pearson's correlation coefficient = 0.112727; Fig. 6b). E-64d treatment caused strong colocalization of the internal SCM-GFP with mCherry-RabF2a (Pearson's correlation coefficient = 0.678813), indicating SCM-GFP accumulation in the prevacuolar compartment (Fig. 6b). We also found that treatment of concanamycin A (ConA) (a vacuolar ATPase inhibitor, which prevents the vacuolar degradation of proteins) greatly increased the accumulation of SCM-GFP in the vacuole in the *qky-16 scm-2* mutant root, while it caused slightly increased vacuolar accumulation of SCM-GFP in the epidermis of the *scm-2* mutant root (Fig. 6c).

Next, we examined the effect of TyrA23, an inhibitor of endocytosis[34], on the SCM-GFP accumulation and found that TyrA23 treatment greatly increased accumulation of SCM-GFP at the PM in the *qky-16 scm-2* mutant root, while it only modestly increased SCM-GFP at the PM in the *scm-2* mutant root (Fig. 6d).

These results indicate that SCM is internalized from the PM by endocytosis and degraded in the vacuole, and that QKY prevents this internalization and vacuolar degradation.

**QKY prevents SCM protein monoubiquitination.** Given the observed size difference between SCM-GFPs from the *SCMp:SCM-GFP scm-2* and the *SCMp:SCM-GFP qky-16 scm-2* roots (Fig. 5b), the accumulation of high-MW SCM-GFP following MG132 treatment (Fig. 5c), and reports showing that ubiquitination is a signal for internalization and degradation of PM proteins[35,36], we examined SCM-GFP for possible ubiquitination. Western blot analysis of immunoprecipitated proteins from

seedlings treated with MG132 using anti-GFP antibodies and anti-ubiquitin antibodies (P4D1) showed that the high-MW SCM-GFP which accumulated following MG132 treatment was ubiquitinated, while the low-MW SCM-GFP protein was not (Fig. 7a). However, antibodies specific for polyubiquitination (FK1) did not react with this high-MW SCM-GFP (Fig. 7a), while both of the antibodies reacted with the positive control BRI1-GFP (which is known to be polyubiquitinated[37]). Considering the homogeneous MW of the ubiquitinated SCM-GFP, this result suggests that the SCM-GFP is monoubiquitinated.

To evaluate the importance of ubiquitination for SCM degradation and QKY's potential role in preventing SCM ubiquitination, we generated an *SCM* construct with all 22 lysine codons in the cytoplasmic domain replaced with arginine codons ($SCM_{m22KR}$). This construct was transiently expressed (*35Sp: $SCM_{m22KR}$-GFP*) in the leaf epidermis of wild-type and *qky-16* mutant plants, and SCM-GFP protein was immunoprecipitated and subjected to western blot analysis (Fig. 7b). Unlike wild-type SCM-GFP, the $SCM_{m22KR}$-GFP did not yield a high-MW SCM-GFP band in the *qky-16* mutant root; rather, the size is similar to the nonubiquitinated form. Indeed, this $SCM_{m22KR}$-GFP did not react with anti-ubiquitin antibodies (P4D1). Furthermore, the fluorescence intensity from the *SCMp:$SCM_{m22KR}$-GFP qky-16 scm-2* plant root was not significantly weaker than the intensity from the *SCMp:$SCM_{m22KR}$-GFP scm-2* plant root unlike the wild-type version of SCM-GFP, which was confirmed by western blot (Fig. 7c). In addition, the difference in the fluorescence intensity between the N-position cells and the H-position cells was much smaller in the *SCMp:$SCM_{m22KR}$-GFP qky-16 scm-2* root and *SCMp:$SCM_{m22KR}$-GFP scm-2* root than in the *SCMp:SCM-GFP scm-2* root (Fig. 7d). Although the fluorescence intensities in the root of those two plant lines were comparable to the *SCMp:SCM-GFP scm-2* line, *SCMp:$SCM_{m22KR}$-GFP* was not able to completely rescue the abnormal root epidermis phenotype of the *scm-2* mutant (Supplementary Fig. 7). This result is reminiscent of the effect of overexpressed SCM in the *scm-2* mutant (*35Sp:SCM scm-2*)[20]. Altogether, these results suggest that monoubiquitination, which is prevented by QKY in the H-position cells, is necessary for differential accumulation of SCM between cells at each position in the root epidermis.

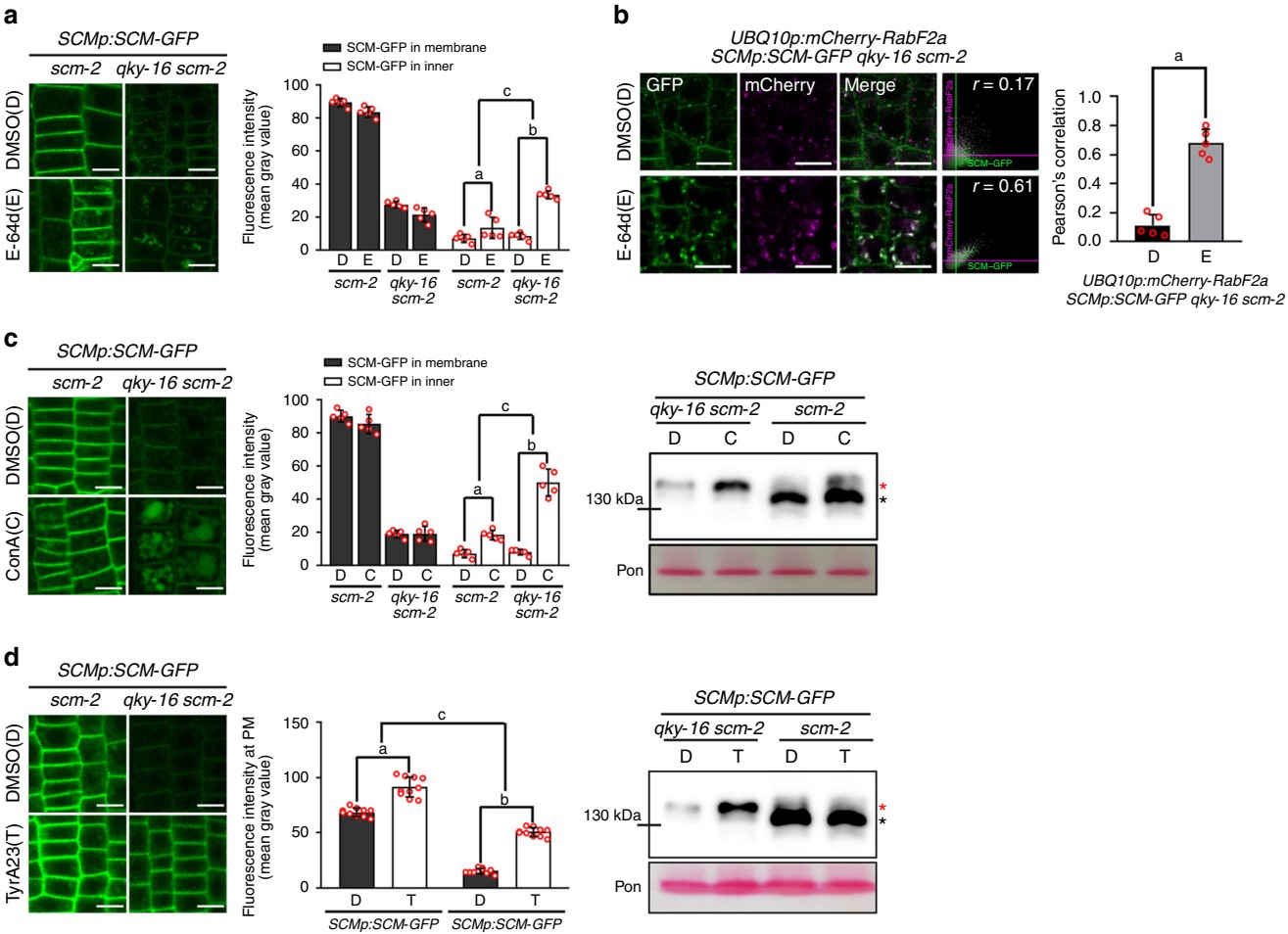

**Fig. 6** The internalization of SCM-GFP is regulated by QKY. **a** Internal accumulation of SCM-GFP after E-64d treatment. Confocal images of the root epidermal cells (left), and the quantified fluorescence intensities in the PM and in the inner area (right) are shown. The seedlings were incubated with or without E-64d (20 μM) for 2 h. *P* values for a, b, and c are 0.033501792889072, 0.000000031378503, and 0.000051222822657, respectively. Bar = 10 μm. **b** Colocalization of internalized SCM-GFP and mCherry-RabF2a. Four-day-old seedlings were treated with or without E-64d (20 μM) for 2 h. The scatterplots show the Pearson's correlation coefficient (*r*) between these two fluorescent signals. Typical examples of images and scatterplots are shown (left). Average values of the Pearson's correlation coefficients from five seedlings are shown in a graph and the error bars indicate standard deviations (right). Statistical significance was determined by unpaired *t* test, and *P* values for a is 0.000006164460956. Bar = 10 μm. **c** Effect of ConA on the SCM-GFP level. Four-day-old seedlings of the two plant lines were incubated with or without ConA (200 nM) for 9 h. Confocal images show internal accumulation of SCM-GFP (left) and the fluorescence intensities in the PM and in the inner area were quantified (middle). The level was also analyzed by western blot (right). *P* values for a, b, and c are 0.002510271018002, 0.000000000221023, and 0.000001097753466, respectively. Bar = 10 μm. **d** Effect of TyrA23 on the SCM-GFP protein level. Four-day-old seedlings of the two plant lines incubated with or without TyrA23 (100 μM) for 10 h. Confocal images show SCM-GFP accumulation (left) and the fluorescence intensity in the PM was quantified (middle). The level was also analyzed by western blot (right). *P* values for a, b, and c are 0.000000000079707, below 0.000000000000001, and 0.000651871734356, respectively. Bar = 10 μm. The asterisks in (**c**) and (**d**) indicate two different bands reacting with anti-GFP antibodies. Statistical significance was determined by two-way ANOVA with Bonferroni's post hoc test (**a**, **c**, **d**). Values represent means ± SD of five individual 4-day-old seedlings for (**a**, **c**) and of ten seedlings for (**d**). In each seedling, ten cells were examined

To identify the ubiquitination sites in the SCM protein, we analyzed the gel mobility of mutant versions of SCM-GFP with single or multiple lysine residues replaced by arginine residues and expressed transiently in *qky-16* mutant leaves (Supplementary Fig. 8). The SCM-GFPs with a substitution of Lys-520, Lys-525 or Lys-526, and two substitutions at Lys-525 and Lys-526 showed intermediate gel mobility (between the mobility of wild-type SCM-GFP in the *qky-16* mutant and the wild type), and SCM-GFPs with two substitutions at Lys-520 and Lys-526 showed similar mobility to that of nonubiquitinated SCM-GFP. Therefore, we changed all three of these lysine residues to alanine (SCM$_{K520/525/526R}$-GFP), expressed this construct transiently in *qky-16* mutant leaves, immunoprecipitated with anti-GFP

antibodies, and analyzed its gel mobility and ubiquitination (Fig. 7e). This SCM$_{K520/525/526R}$-GFP showed similar mobility to that of nonubiquitinated SCM-GFP and did not react with anti-ubiquitin antibodies (P4D1) (Fig. 7e). These results suggest that SCM is modified by multi-monoubiquitination and that these three residues (Lys-520, and at least one from Lys-525 and Lys-526) represent the ubiquitination site.

## Discussion

SCM, an atypical LRR RLK, is a critical regulator of cell-type patterning in the *Arabidopsis* root epidermis[16]. However, the mechanism and regulation of SCM action had been unclear prior

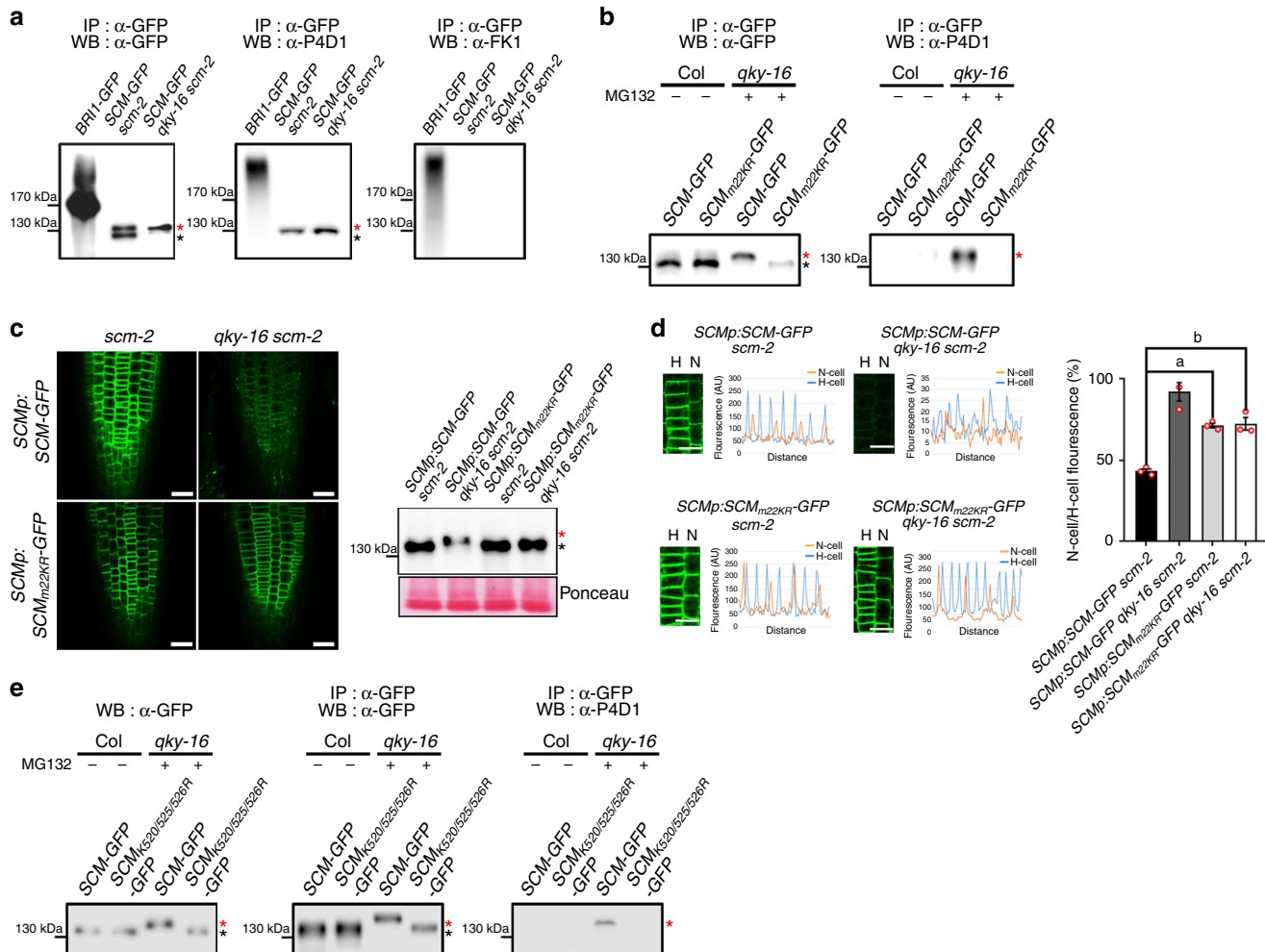

**Fig. 7** QKY regulates the ubiquitination of SCM. **a** In vivo ubiquitination analysis of SCM-GFP. Immunoprecipitation was performed on protein extracted from MG132-treated (20 μM) *SCMp:SCM-GFP scm-2* and *SCMp:SCM-GFP qky-16 scm-2* seedlings and the resulting immunoprecipitates were subjected to immunoblotting with monoclonal anti-GFP (left), anti-Ub P4D1 (middle) and anti-polyUb FK1 (right). **b** In vivo ubiquitination analysis of SCM-GFP and SCM$_{m22KR}$-GFP. Ubiquitination assay was performed as in (**a**) with the immunoprecipitates from *Arabidopsis* leaves transiently expressing *SCM-GFP* or *SCM$_{m22KR}$-GFP* for 12 h with or without MG132 (20 μM) after introducing *35Sp:SCM-GFP* or *35Sp:SCM$_{m22KR}$-GFP* by the Biolistic PDS-1000/He system. **c** GFP accumulation in the root epidermal cells of *SCMp:SCM-GFP scm-2*, *SCMp:SCM-GFP qky-16 scm-2*, *SCMp:SCM$_{m22KR}$-GFP scm-2* and *SCMp:SCM$_{m22KR}$-GFP qky-16 scm-2* (left). SCM$_{m22KR}$ indicates that all the lysine residues in the cytoplasmic domain of SCM protein were mutated to arginine residue. Same confocal detection setting was used to compare the fluorescence intensities from different plants. Bar = 25 μm. The protein level was also analyzed by western blot (right). **d** Aberrant differential accumulation pattern of SCMm$_{22KR}$-GFP between the N-position cells and the H-position cells. Fluorescence intensities were measured in two adjacent cell files from the starting point of elongation zones (left) and the intensities of the four peaks in the N-position cells and the nearest four peaks in the H-position cells were compared (right). Error bars indicate standard deviation from three independent transgenic lines. Five plants were analyzed in each transgenic line. Statistical significance was determined by one-way ANOVA with Bonferroni's post hoc test, and the *P* values for a and b are 0.003198883401511 and 0.002495323713490, respectively. *AU* arbitrary unit, Bar = 25 μm. **e** Identification of ubiquitination sites in SCM protein. Ubiquitination assays were performed as in (**a**) with the immunoprecipitates from *Arabidopsis* leaves transiently expressing SCM-GFP or SCM$_{K520/525/526R}$-GFP in which lysine residues at 520, 525, and 526 were substituted by arginine for 12 h with or without MG132 (20 μM) after introducing *35Sp:SCM-GFP* or *35Sp:SCM$_{K520/525/526R}$-GFP* by the Biolistic PDS-1000/He system. In (**a**), (**b**), and (**e**), the red asterisk indicates ubiquitinated SCM-GFP and black asterisk indicates nonubiquitinated SCM-GFP

to this study. Here, we showed that SCM function in root epidermal patterning is dependent on QKY. Like *scm* mutants, the *qky* mutant lacked position-dependent cell-type specification (Supplementary Fig. 1). The *QKY* gene displayed a similar root expression pattern as the *SCM* gene, and the QKY and SCM proteins both accumulated in the PM, preferentially in the H-position cells (Fig. 1). Furthermore, we found that *SCM* and *QKY* are likely to act in the same genetic pathway (Fig. 2).

We have also shown that QKY and SCM are required for movement and preferential accumulation of CPC in the H-

position epidermal cells (Fig. 3). PD are intercellular channels conferring cytoplasmic continuity between plant cells and controlling movement of various molecules by targeted and nontargeted mechanisms. Some plant RLKs including SCM appear to associate with PD[38–41], and some of these are suggested to regulate movement through PD[39,40]. Interestingly, a QKY-related protein (one of the 16 members of the multiple C2 domain and transmembrane region proteins (MCTP) family in *Arabidopsis*) called FT-INTERACTING PROTEIN1 (FTIP1) shows preferential localization to PD and regulation of specific protein

(FLOWERING LOCUS T (FT)) movement between cells[42]. FTIP was shown to directly interact with FT in companion cells through its third C2 domain and to control FT export from companion cells to the neighboring sieve elements[42]. In contrast, we discovered that QKY directly interacts with SCM through its N-terminal region containing C2 domains (Fig. 4) rather than CPC (Supplementary Fig. 3) and seems to control CPC import rather than export between epidermal cells (Fig. 3c). Furthermore, QKY seems to regulate the movement of CPC indirectly, as QKY is a regulator of SCM protein accumulation (Figs. 5 and 6).

Our discovery of QKY-dependent stabilization of the SCM protein provides an explanation for the close relationship between QKY and SCM. We show that QKY directly interacts with SCM and generates high levels of nonubiquitinated SCM, rather than ubiquitinated SCM which is internalized and degraded in the vacuole (Figs. 4–6). The precise mechanism by which QKY generates nonubiquitinated SCM is unclear. QKY may promote SCM deubiquitination or it may prevent SCM ubiquitination. The former possibility appears unlikely, because although QKY directly interacts with SCM through their extracellular domains (Fig. 4), QKY has no discernible protease domain nor shows any similarity to known deubiquitinating enzymes. The effect of QKY on SCM stability may also explain the previously reported preferential accumulation of SCM in the H-position cells of the root epidermis[20]. Notably, a nonubiquitinated mutant version of SCM ($SCM_{m22KR}$) expressed under the control of the SCM promoter accumulated highly in both epidermal cell positions (H cell and N cell), even in the absence of QKY (Fig. 7c, d). This implies that the observed preferential H-cell accumulation of wild-type SCM is due to preferential SCM stabilization in the H-cell position as a result of the greater QKY accumulation in the H-position cells. Accordingly, an important future goal is to explore the mechanism responsible for preferential H-position cell accumulation of QKY.

In mammals and yeast, monoubiquitination and K63-polyubqubination generally govern trafficking of membrane proteins, whereas K48-polyubiquitination generally regulates proteasome-dependent protein degradation[43,44]. In plants, diverse forms of ubiquitination appear to be used for the degradation of different membrane proteins. Polyubiquitination has been reported to be important for internalization and degradation of LRR RLKs, such as BRI1 and FLAGELLIN SENSING2 (FLS2)[37,45]. Other membrane proteins have also been reported to be ubiquitinated. The multi-monoubiquitination of ION-REGULATED TRANSPORTER 1 (IRT1) drives its internalization to early endosomes (EE)[46] and subsequent K63-polyubiquitination of the multi-monoubiquitin moieties is required for its vacuolar degradaion[47]. BOR1, a boron transporter protein, is mono- or diubiquitinated and degraded in the vacuole[48]. We found that SCM is multi-monoubiquitinated, and this promotes its internalization and vacuolar degradation (Figs. 6c, d and 7). Taken together, mono-ubiquitination seems to be sufficient for the internalization of membrane proteins in plants, as observed in nonplant organisms[43,44], but multi-monoubiquitination and K63-polyubiquitination may increase the efficiency of internalization, as also shown in yeast[49,50], and may be responsible for the vacuolar targeting. It is possible that SCM is also polyubiquitinated, which may have gone undetected due to its low level. Here, however, we showed that blocking SCM degradation in vacuoles resulted in greater accumulation of multi-monoubiquitinated SCM, implying that multi-monoubiquitination plays a major role in the vacuolar degradation of SCM even though we cannot rule out the possibility of polyubiquitination.

Together, these results provide an updated model for cell fate specification in the Arabidopsis root epidermis. In this model, QKY protein preferentially accumulates in the H-position cells of

the root epidermis and stabilizes SCM protein in these cells by preventing its ubiquitination. The preferential accumulation of SCM at PD in the H-position cells facilitates import of CPC protein from the neighboring N-position cells. The accumulation of CPC in the H-position cells inhibits WER function[9]. In turn, feedback regulation (negative regulation by WER and positive regulation by CPC on SCM expression[20]) further establishes preferential accumulation of SCM protein in H-position cells, which ultimately results in the cell-type pattern. This model is consistent with the inability of QKY overexpression to rescue the scm-2 mutant phenotype (Table 1) and the report that increasing SCM gene copy number shows a mild reduction of the qky mutant phenotype[23]. It is also consistent with the finding that the scm mutation and the qky mutation did not cause a significant increase in hair cell specification in the N-position cells in the absence of CPC (Table 1)[21]. However, we also found that ubiquitous expression of QKY in the wild type did not cause ectopic hair cell fate specification in the N-position cells (Table 1), which indicates that an additional H-position factor might be required for effective stabilization of SCM by QKY.

## Methods

**Plant materials and growth conditions.** The wer-1, cpc-1, scm-2, and qky-8 alleles were described previously[7,8,16,22]. The GL2p:GUS, WERp:GFP, CPCp:GUS, EGL3p:GUS, SCMp:GUS, SCMp:SCM-GFP and SHRp:CPC-GFP reporter gene lines were also described previously[6,7,13,16,20,25,51].

For plant growth, seeds were sterilized, germinated and grown vertically on agarose-solidified medium containing mineral nutrients[52].

**Analysis of epidermal cell patterning.** For each line, we analyzed at least ten 4-day-old seedlings and performed three independent experimental repeats. We scored the ten H-position cells and the ten N-position cells for each seedling using differential interference contrast microscopy.

**Gene constructs and plant transformation.** For qky-16 complementation experiments, the genomic DNA fragment containing a 1210-bp 5′ flanking region DNA fragment and a 576-bp 3′ flanking region DNA fragment from QKY were used. To analyze the promoter activity of QKY, the β-glucuronidase gene or the GFP gene was inserted between the same 5′ and 3′ flanking DNA fragments that were used for the complementation test (QKYp:GUS and QKYp:GFP). To visualize the QKY protein, the GFP coding region was inserted at the end of the first transmembrane domain of QKY (QKY-GFP). For the QKYp:QKY-GFP construct, the chimeric QKY-GFP was inserted between the same flanking sequences of QKY. To express QKY-GFP in various tissues, the chimeric gene was fused to each tissue-specific promoter. For the RHD3p:QKY construct, QKY was fused to the 3′ end of the RHD3 promoter[30]. $SCM_{Em1}$-GFP and $SCM_{Em2}$-GFP fusions were fused to the 3′ end of the SCM promoter[20]. For the BiFC assay, QKY-EYFPc-V1, QKY-EYFPc-V2, QKY-EYFPc-V3, SCM-EYFPn-V1 and SCM-EYFPn-V2 constructs were made by PCR cloning using Gibson assembly technology and expressed under the control of the 35S promoter. To identify ubiquitination sites, SCM, $SCMm_{22KR}$ and $SCMm_{K520/525/526R}$ were fused to 5′ end of the GFP gene, and transiently expressed under the control of the 35S promoter in Arabidopsis leaves. Plant transformation was achieved by electroporating constructs into the Agrobacterium strain GV3101 followed by introduction into Arabidopsis using the floral dip method[53]. The primers used for the constructions are described in Supplementary Table 1.

**Chemical treatments.** Inhibitors (Sigma-Aldrich) were used at the following concentrations: 200 nM ConcA, 100 μM TyrA23, 20 μM E-64d, 50 μM MG132, 50 μM lactacystin and 10 ng/μl proteinase K.

**Histochemical GUS staining.** GUS activity was examined by observing 4-day-old seedlings stained for 1−12 h in a solution containing 5-bromo-4-chloro-3-indolyl glucuronide[12].

**Confocal microscopy.** To analyze GFP expression, seedlings were counterstained with 5 μg/ml propidium iodide (PI) or 5 μM FM4-64 (Invitrogen) and examined using a Zeiss LSM880 confocal microscope[7]. The PI signal was detected with a 543 nm excitation mirror and a 560 nm long-path filter. The mCherry-RabF2a signal was detected with a 561 nm excitation mirror and a 661 nm long-path filter. Laser intensity settings were kept constant across samples in each experiment.

**Transient gene expression with the particle gun.** Rosette leaves were bombarded with DNA-coated tungsten particles using the Biolistic PDS-1000/He system

(Bio-Rad) according to the manufacturer's instructions. Tungsten (M10) particles were coated with 1 μg of the DNA, or with 1 μg of each DNA for co-bombardment. The analysis was carried out after 4−12 h of incubation.

**Yeast two-hybrid assay.** The C2-domain containing region of QKY (QKY$_{D1}$), second domain of QKY (QKY$_{D2}$), and third domain of QKY (QKY$_{D3}$) were joined as a C-terminal fusion to the yeast GAL4 DNA-binding domain in pAS2-1. Likewise, the extracellular domain (SCM$_{ED}$), the cytosolic domain (SCM$_{CD}$) of SCM, and the mutated extracellular domains (SCM$_{Em1}$, SCM$_{Em2}$) were fused to the GAL4 activation domain in pACT2. Transformation was performed using yeast strain AH109 and assays were examined in selective media. The gene-specific primers used are listed in Supplementary Table 1.

**Immunoprecipitation and western blot analysis.** Total proteins were extracted from 4-day-old seedlings or transiently expressed rosette leaves. For western blot analyses, total proteins were extracted from approximately 100 mg of tissue. The following antibodies were used in these experiments: monoclonal anti-GFP (Clontech 632380,1/3000), polyclonal anti-GFP (abcam ab290, 1/2000), anti-FLAG (Abm G191, 1/3000), anti-cMYC (Calbiochem OP10L, 1/3000), anti-ubiquitin P4D1 (Millipore 05-944, 1/2500) and anti-polyubiquitin FK1 (Enzolifesciences BML-PW8805-0500, 1:1000). Uncropped blots are shown in Source Data file. One gram of seedlings or transiently expressed rosette leaves was used for the immunoprecipitation experiments. The tissue was ground in liquid nitrogen, resuspended in 1 ml of immunoprecipitation buffer (50 mM Tris pH 7.5, 150 mM NaCl, 1% Triton X-100 and protease inhibitor cocktail) and left on a rotating wheel for 30 min at 4 °C. Samples were then centrifuged for 10 min at 20,000 × g at 4 °C. Immunoprecipitations were carried out on 1 mg of total proteins using the EZview™ Red ANTI-FLAG® M2 Affinity Gel (Sigma) and GFP-Trap-A (Chromotek) according to the manufacturer's protocol.

**Quantitative real-time RT-PCR.** To measure the steady-state level of transcripts, total RNA was extracted from the root tips of 4-day-old seedlings and treated with RNase-free DNase I. One microgram of total RNA was used for the reverse transcription using the AccuScript High Fidelity 1st Strand cDNA synthesis kit (Agilent Technologies, 200820), and 1 μl of resulting first-strand cDNA (20 μl) was used as a PCR template for the quantitative real-time RT-PCR. Quantitative PCR analysis was performed using SYBR Premix EX Taq (Takara, #RR82LR) with an AriaMx real-time PCR machine (Agilent Technologies). *EF1* was used as an internal reference to normalize the relative level of each transcript. Each experiment was repeated three times, and each time the experiment included triplicate samples. The raw data underlying the averages are shown in Source Data file.

**Quantification of SCM-GFP accumulation.** Confocal images were obtained from ten 4-day-old seedlings of the corresponding plant line. Mean fluorescence intensities of SCM-GFP at the PM and in the inner area were measured in the ten H-position epidermal cells from each seedling image by creating a region of interest covering the PM area and entire cytoplasm area, respectively using ImageJ program (Fiji version). The raw data underlying the averages are shown in Source Data file.

**Analysis of colocalization of internalized SCM and RabF2a.** Confocal images of 4-day-old seedling roots were analyzed using the Image-pro 10 software to produce scatterplots and to calculate colocalization of SCM-GFP and mCherry-RabF2a. The scatterplots that were obtained by analyzing all pixels inside five different cells in one seedling produced the Pearson's correlation coefficient based on the automated Otsu threshold[54]. The raw data underlying the averages of the Pearson's correlation coefficient are shown in Source Data file.

**Proteinase K protection test.** Protoplasts were isolated from *SCMp:SCM-GFP* and *QKYp:QKY-GFP* stable transgenic plants. The protoplasts were incubated in the presence or absence of proteinase K (1 ng/μl). Then, PMSF and SDS sample buffer were added and denatured at 95 °C to inactivate protease activity.

**Bimolecular fluorescence complementation.** Fifty microliters of *Agrobacterium* overnight culture was centrifuged, washed, and suspended in a solution containing 10 mM MES, 10 mM MgCl$_2$, and 200 μM acetosyringone. Coinfiltration of *Agrobacterium* strains containing the BiFC constructs and the p19 silencing plasmid was carried out. The leaves of 4-week-old *Nicotiana benthamiana* were infiltrated and subsequently analyzed at day 3 after infiltration using a Zeiss LSM 880 confocal microscope with ZEN software. EYFP signal was detected with a 514 nm excitation mirror and a 520−600 nm band-path filter.

**Reporting summary.** Further information on experimental design is available in the Nature Research Reporting Summary linked to this article.

## Data availability

The source data for Figs. 3−7, Supplementary Figs. 4−6, 8, and Table 1 are provided as a Source Data file. All other data that support the findings of this study are available from the corresponding author upon reasonable request.

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

## Acknowledgements
This work was supported by a grant from Woo Jang Choon Project (grant number, PJ01093905 to M.M.L.) of the Rural Development Administration (RDA), Republic of Korea, and by a grant from the Basic Science Research Program through the National Research Foundation (NRF) (grant number 2018R1A6A1A03025607) funded by the Ministry of Education, Republic of Korea.

## Author contributions
J.H.S. contributed in designing and performing the experiments related to the expression of *QKY* and *SCM*, interaction between QKY and SCM, CPC movement, SCM ubiquitination. S.-H.K. contributed in designing and performing the experiments related to the expression of *QKY* and *SCM*. K.H.N. contributed in designing and performing the Co-IP experiments. J.S. and M.M.L. designed the experiments and wrote the manuscript.

## Additional information

**Competing interests:** The authors declare no competing interests.

