## [Peer Review File · Nature Communications]

Reviewers' comments:

Reviewer #1 (Remarks to the Author):

The manuscript of Song et al entitled "QUIRKY regulated root epidermal cell patterning through stabilizing SCRAMBLED to control CAPRICE movement in Arabidopsis" reports that the QUIRKY (QKY) regulates the ubiquitin-mediated vacuolar degradation of SCRAMBLED (SCM) by interaction and that QKY and SCM are both important for the cell to cell movement of the cell fate regulator CAPRICE (CPC). Despite intensive studies, the molecular mechanisms of cell fate specification during plant development has not yet been completely understood. Fate determination of root hair vs non-root hair cells is an excellent model to study cell fate determination as a number of specific markers and regulators have been established in the past years.

With the aim to identify factors that regulate cell fate patterning in the epidermis, the authors screened for EMS-mutants that show altered expression pattern of the GL2p:GFP reporter that specifies non-root hair cells. The authors identified a novel mutant allele of *qky* named *qky-16* in this screen that showed epidermal patterning defects. Further genetic analysis suggested that QKY and SCM act in the same pathway and that QKY is upstream of SCM. To establish whether the mobility of the mobile transcription factor CPC is affected by QKY and SCM, the authors next analyzed the radial and lateral movement of CPC in mutant backgrounds. The results show that the radial movement as well as cell export of CPC-GFP remains unaffected in *qky* and *scm* mutants, implying that QKY and SCM are important in the import and accumulation of CPC in root hair cells. Protein-protein interaction studies revealed that QKY and SCM interact by their extracellular domains. The authors observed that the level of SCM-GFP is reduced in a *qky* mutant background and that the mobility of SCM-GFP on a protein gel is reduced. The authors show that this band corresponds to multi-monoubiquitinated SCM-GFP and suggests that QKY stabilizes SCM through physical interaction possibly by inhibition of ubiquitination and degradation of SCM. By this mechanisms, QKY could regulate the specific accumulation of SCM in H-cells, which could be important for the specific recruitment and expression of further transcriptional regulators.

Most of the data presented in the ms are of high quality and supports the conclusion of the authors. The work presents a novel possible regulatory mechanism of epidermal cell patterning and has the potential to be appreciated by the general readership. Though the authors nicely show that QKY and SCM interact and this interaction might be important for regulating the stability and thus the accumulation of SCM in the H-position, at this stage, this reviewer has several concerns regarding some of the experiments as listed below.

Major comments

1. Figure 3e and text L278-280: "In addition, the fluorescence intensity from the SCME₂-GFP which fails to interact with QKY was much weaker than the intensity from SCM-GFP and from SCME₁-GFP even though they all were expressed under the control of the SCM promoter and in the same *scm-2* genetic background."

Even when the constructs use the same promoter and are in the same genetic background, the expression level of the wild-type and mutant protein might differ. To exclude that the "weaker" GFP signals is indeed due to posttranscriptional regulation, expression of the SCM-GFP variants should be tested in a qRT-PCR and the protein amounts analyzed on a Western blot. Especially, the Western blot would be essential to prove that the interaction between SCM and QKY affects the ubiquitination status of SCM and as a consequence its abundance.

2. Figure 4a: As mentioned for Figure 3e, the transcript level of SCM-GFP should be analyzed by qRT-PCR to exclude the possibility that the transgene behaves differently.

3. Figure 4d: What are the structures stained with FM4-64 after E-64d treatment? It is not clear whether the seedlings were only transiently treated with FM4-64 or continuously supplied with the dye. To convincingly show that SCM-GFP traffic occurs via the endocytosis route, the authors

might want to consider conducting a time course experiment with FM4-64 and monitor the colocalization of SCM-GFP with the dye over time. This would be a very important experiment, since the authors do not use any endomembrane/endosome markers to investigate the intracellular localization of SCM-GFP.

4. Figure 4f: TyrA23 was recently shown not to be a CME inhibitor (Dejonghe et al. 2015 Nature Communications) and thus the results of this experiment does not support the conclusion of the authors that SCM-GFP degradation is CME-dependent.

5. Fig 5 and text L 433-437 "...Taken together, monoubiquitination seems to be sufficient for the internalization of membrane proteins in plants, as observed in non-plant organisms": To this reviewer's knowledge, the current understanding is that membrane proteins need to be Lys63-polyubiquitinated in order to enter the lysosomal/vacuolar degradation pathway and monoubiquitination or multi-monoubiquitination is not sufficient (also in plants). In the example of IRT1 that the authors cite (L433 and 443), multi-monoubiquitination was shown to be not sufficient for the degradation of IRT1 in the vacuole and IRT1 is transported to the vacuole for degradation only upon Lys63-polyubiquitination by the E3 ligase IDF1 (Dubeaux et al. 2018 Mol Cell). Would a longer exposure of the anti-ubiquitin Western blot or increase in material show polyubiquitination of SCM-GFP?

Reviewer #2 (Remarks to the Author):

The manuscript titled "QUIRKY regulates root epidermal cell patterning through stabilizing SCRAMBLED to control CAPRICE movement in Arabidopsis" reported interesting findings of the role of QKY in cellular patterning of Arabidopsis epidermis and the involvement of QKY in stabilizing SCM. In addition to routine characterization of the qky mutant, the authors provided detail analysis of interaction between QKY and SCM. They clarified the interacting region between QKY and SCM, demonstrated the requirement of QKY in SCM stabilization and SCM monoubiquitination. These findings provided very valuable information in the efforts of further deciphering the mysterious role of SCM in the cellular patterning of Arabidopsis root epidermis.

It is obvious that the authors made great efforts to analysis of the mechanism of interaction between QKY and SCM. However, it is crucial to show the colocalization of the two protein in vivo. At least two methods could be useful for the purpose, either BiFC or more precisely using different fluorescent proteins to localize the two protein respectively and merge the imagines. Probably there were some difficulties in practice. However, the authors have made efforts to insert the GFP into the two domains to obtain a functional construct, if they want, it would be no doubt that they can accomplish such a colocalization assay.

It is very interesting finding that QKY can affect differential distribution of SCM in H/N cells. In this work, authors provided different lines of evidence to demonstrate the role of QKY in SCM ubiquitination, degradation and distribution. However, it is still not clear how these interactions at protein level can alter the SCM differential distribution in epidermal cells. In the discussion, the authors mentioned the abundant regulation in protein distribution. In their data, there were some clues that the ratio of mono vs multi-ubiquitination or the rate of movement to vacuole degradation might contribute the abundant regulation of SCM distribution. However, additional to the role of the change in different genetic background with genetic data, there are no direct evidences at cellular level to support the speculation.

The third issue concerned is about the ubiquitination sites. The authors demonstrated that at the

interaction region between QKY and SCM, there are three ubiquitination sites. After all of three substituted, the function of SCM was lost. It is reasonably asking whether the all three amino acids required or only one or two of them are required? Any author hate to be asked to add more data. However, it is for sure that the authors are doing experiments to the questions as such question are too obvious to be addressed.

Following the above concerns or comments, this manuscript read loosely organized. The part of mutant characterization is necessary for the work and well organized of the first three sessions of results. But the following parts seems diffused. Several issues were addressed, such as CPC movement, PD limitation and protein abundance regulation. None of these issues seems effectively investigated. If to focus the mechanistic investigation into the interaction between the QKY and SCM and propose one or two definitive conclusions on the role of QKY on SCM distribution, this work would be presented more coherently and readably

The minor issue is the organization of writing of discussion part. It can be significantly shorten and concisely organized.

Reviewer #3 (Remarks to the Author):

In this manuscript Song et al. demonstrated that QKY regulates epidermal cell patterning by affecting the expression of epidermal cell fate regulators and additionally the movement of CPC. They proved that QKY interact directly with SCM by extracellular domain rather than the previous reported C2 domain to prevent SCM degradation and regulate CPC movement. As we know still relatively little about factors that contribute to protein movement, I think the current study is interesting. However, at this stage I have two general comments.

First, although the authors focus on especially the root epidermal, the CPC movement assays are done in leaf cells. Why didn't the authors perform the movement assays in the physiological context? Is this because of a technical reason?

My second major concern is most important analyses in this work lack quantification data, which are pointed out in the following comments:

1. Line 158. Figure 1b and the statement that QKY seems to be more abundant in H cells is not acceptable without some form of quantification. An optical section image is a better choice. Arrows should also be provided to guide the reader to different cells. Figure 1c, Figure 2a, 3a lack the quantification. I appreciate that it is in Table 1 but I don't think it is a great way to represent data. I think they should be in the same figure as a histogram, much better for visualisation.

2. The authors claimed "the CPC-GFP did not exhibit greater accumulation in the H position cells in qky-16 and scm-2" in Line 202. It is rather fundamental for their argument so strong evidence are needed for support. Images of Fig2b is not convincing for the conclusion without quantification. Also, the data in leaves shown in Fig2c makes their arguments somewhat weaker as the authors was trying to explore the function of QKY in root, not in leaves. What kind of movement is then being suggested? GFP movement not affected, CPC-GFP affected. Are they speculating a facilitated movement with chaperones helping the passage of CPC? How does that relate to QKY ecc. How would they then reconcile this with the speculation that QKY involved in the import, rather than export of CPC? I find a gap and some confusion here in the manuscript about the actual mechanism of movement.

3. While analyzing SCM translation level, the authors using transgenic lines with SCMp::SCM-GFP in qky-16scm-2 and scm-2 background. They claimed the SCM promoter activity and SCM-GFP transcript level are similar in the two background (supp. Fig5). However, they did not show

quantification of GUS and SCM-GFP transcript level in those transgenic lines, which is necessary for comparing SCM protein levels.

4. Figure 4a the first image is clearly saturated. I would prefer a quantification for panel a,d, e, f (all the microscopy). I think that should be the standard these days for image analysis. That said I appreciate that having protein levels next to these confocal images is somewhat of a form of quantification. Similar argument for Fig 5c.

5. In western blotting assay, there were two protein bands reacting with anti-GFP antibody, which was explained by ubiquitination and reduced gel mobility. I would suggest the authors to confirm it is SCM-GFP using another GFP antibody and SCM antibody.

6. All the data they have about the mechanism of internalisation are based on chemical treatments; I wonder if they could think of potential further evidence of a different nature?

Response to reviewers

My coauthors and I have received and read the comments from the reviewers regarding our submitted manuscript “QUIRKY regulates root epidermal cell patterning through stabilizing SCRAMBLED to control CAPRICE movement in Arabidopsis” (manuscript number, NCOMMS-18-29174). We appreciate the helpful advice and suggestions, and we have modified our manuscript accordingly to make the enclosed revised manuscript. Our detailed response to the reviewers' comments follows.

<Reviewer 1>

Comment 1: Figure 3e and text L278-280: “In addition, the fluorescence intensity from the SCME_{m2}-GFP which fails to interact with QKY was much weaker than the intensity from SCM-GFP and from SCME_{m1}-GFP even though they all were expressed under the control of the SCM promoter and in the same scm-2 genetic background.”

Even when the constructs use the same promoter and are in the same genetic background, the expression level of the wild-type and mutant protein might differ. To exclude that the “weaker” GFP signals is indeed due to posttranscriptional regulation, expression of the SCM-GFP variants should be tested in a qRT-PCR and the protein amounts analyzed on a Western blot. Especially, the Western blot would be essential to prove that the interaction between SCM and QKY affects the ubiquitination status of SCM and as a consequence its abundance.

Response: (Supplementary Fig. 6; Line 271-274) We agree with this reviewer's comment. Accordingly, we have performed and included new qRT-PCR and Western blot experiments as the reviewer suggested. The qRT-PCR results show that the steady-state levels of the recombinant GFP transcripts in the roots of the *SCMp:SCM-GFP scm-2*, the *SCMp:SCM_{Em1}-GFP scm-2*, and the *SCMp:SCM_{Em2}-GFP scm-2* are similar, while the Western blot results

show a reduced accumulation of the recombinant GFP protein in the roots of the *SCMp:SCM_{Em2}-GFP scm-2* than in the *SCMp:SCM-GFP scm-2* and the *SCMp:SCM_{Em1}-GFP scm-2* (Supplementary Fig. 6). These results indicate that the difference in the protein levels was not caused by a difference in promoter activities nor by a difference in transcript stability. Now, this is described in the results section (line 271-274). Furthermore, we found that anti-GFP antibodies reacted with a higher MW protein band in the *SCMp:SCM_{Em2}-GFP scm-2* compared with the protein in the *SCMp:SCM-GFP scm-2* and the *SCMp:SCM_{Em1}-GFP scm-2*, as the reviewer expected. Based on these new results, we are able to suggest more convincingly that the interaction between QKY and SCM is important for the stabilization of SCM.

Comment 2: Figure 4a: As mentioned for Figure 3e, the transcript level of SCM-GFP should be analyzed by qRT-PCR to exclude the possibility that the transgene behaves differently.

Response: (Supplementary Fig. 5a and Supplementary Fig. 5c; Line 268-271)

In this case, we actually used the same transgenic line (*SCMp:SCM-GFP*) to make the *SCMp:SCM-GFP scm-2* plant and the *SCMp:SCM-GFP qky-16 scm-2* plants (via genetic crosses). Furthermore, we showed that the promoter activity of SCM in the *qky-16* root was not significantly different from its activity in the wild-type root based on the *SCMp:GUS* construct activity, which was introduced into these mutants by genetic crosses (Supplementary Fig. 5a). In addition, we also showed that the steady state level of *SCM-GFP* transcript in the the *SCMp:SCM-GFP scm-2* mutant root was not different from the level in the *SCMp:SCM-GFP qky-16 scm-2* mutant root (Supplementary Fig. 5c). We have described this in the results section (line 268-271) of the revised manuscript. Together, these results enable us to exclude the possibility that the transgene behaves differently in these two lines.

Comment 3: Figure 4d: What are the structures stained with FM4-64 after E-64d treatment? It is not clear whether the seedlings were only transiently treated with FM4-64 or continuously supplied with the dye. To convincingly show that SCM-GFP traffic occurs via the endocytosis route, the authors might want to consider conducting a time course experiment with FM4-64 and monitor the colocalization of SCM-GFP with the dye over time. This would be a very important experiment, since the authors do not use any endomembrane/endosome markers to investigate the intracellular localization of SCM-GFP.

Response: (Fig. 5e; Line 289-295) The FM4-64 was treated with or without E-64d for 2 h. Therefore, most of the FM4-64 inside the cell likely accumulated in late endosomes.

To more clearly show the localization of SCM-GFP after the E-64d treatment, we have made the *qky-16 scm-2* mutant harboring the *SCMp:SCM-GFP* and the *UBQ10p:mcherry-RabF2a* (a late endosome marker) (Geldner et al., 2009) by a genetic cross. This new result shows that the SCM-GFP protein accumulates in late endosomes after the treatment of E-64d, which indicates that SCM-GFP is targeted to the vacuole in the absence of E-64d treatment. We have included this result in Fig. 5e and described it on the results section (line 289-295).

Comment 4: Figure 4f: TyrA23 was recently shown not to be a CME inhibitor (Dejonghe et al. 2015 Nature Communications) and thus the results of this experiment does not support the conclusion of the authors that SCM-GFP degradation is CME-dependent.

Response: (Line 300-305) As the reviewer points out, although TyrA23 is widely used as a CME inhibitor, it has been shown that it induces cytoplasmic acidification which indirectly inhibits CME rather than directly affecting the function of a CME component. To more effectively demonstrate that SCM internalization is CME-dependent, it would be necessary to use an alternative method, such as an inducible clathrin loss-of-function line (e.g. inducible

dominant-negative HUB1). However, the main point in our experiment was to show that SCM is rapidly internalized from the plasma membrane in the absence of QKY rather than whether it's internalization is CME-dependent or not. Therefore, in response to this comment, we have modified the text to describe this effect more generally as endocytosis instead of CME-dependent endocytosis (line 300-305).

Comment 5: 5. Fig 5 and text L 433-437 “....Taken together, monoubiquitination seems to be sufficient for the internalization of membrane proteins in plants, as observed in non-plant organisms”: To this reviewer's knowledge, the current understanding is that membrane proteins need to be Lys63-polyubiquitinated in order to enter the lysosomal/vacuolar degradation pathway and monoubiquitination or multi-monoubiquitination is not sufficient (also in plants). In the example of IRT1 that the authors cite (L433 and 443), multi-monoubiquitination was shown to be not sufficient for the degradation of IRT1 in the vacuole and IRT1 is transported to the vacuole for degradation only upon Lys63-polyubiquitination by the E3 ligase IDF1 (Dubeaux et al. 2018 Mol Cell). Would a longer exposure of the anti-ubiquitin Western blot or increase in material show polyubiquitination of SCM-GFP?

Response: (figure 1 below; Line 406-409 and Line 415-422) We appreciate this reviewer's comment. We had overlooked the Dubeaux's report (2018) about the K63-polyubiquitination of IRT1. As the reviewer points out, this group showed that multi-monoubiquitination drives IRT1 internalization to early endosomes and K63-polyubiquitination is required for its vacuolar degradation. We have changed the description on the IRT1 ubiquitination in line 406-409. As the reviewer suggested, we have already tried to expose much longer time and load more immunoprecipitated proteins in Western blot analysis. Infrequently, we detected faint smear signals over the major band, but generally we were not able to detect consistent signal (see the figure 1 below). We are also unsure whether the faint smear signals are increased in the *qky scm* mutant root after MG132 treatment compared with the *scm*

mutant root (due to its low level and inconsistency). However, we were able to detect the multi-monoubiquitinated SCM-GFP consistently in the *qky scm* mutant and even in the *scm* mutant after MG132 treatment. We also show that blocking SCM degradation in vacuoles by ConA treatment resulted in greater accumulation of multi-monoubiquitinated SCM, implying that multi-monoubiquitination plays a major role in the vacuolar degradation of SCM, even though we cannot rule out the possibility that we were not able to detect polyubiquitination of SCM due to its low level and that it still plays a role in SCM degradation. To make this clear, we have modified the description about this in the discussion section (line 415-422).

Figure 1. *In vivo* ubiquitination analysis of SCM-GFP. Immunoprecipitation was performed on protein extracted from MG132-treated (20 μ M) *SCMp:SCM-GFP scm-2* and *SCMp:SCM-GFP qky-16 scm-2* seedlings, and the resulting immunoprecipitates were subjected to immunoblotting with anti-GFP and anti-polyUb FK1 antibodies. **a.** Infrequently, western blot analysis using anti-polyUb FK1 antibodies showed reacting bands after a longer exposure. **b.** Generally, western blot analysis using anti-polyUb FK1 antibodies showed no detectable reacting bands even after a longer exposure.

<Reviewer 2>

Comment 1: It is obvious that the authors made great efforts to analysis of the mechanism of interaction between QKY and SCM. However, it is crucial to show the colocalization of the two protein in vivo. At least two methods could be useful for the purpose, either BiFC or more precisely using different fluorescent proteins to localize the two protein respectively and merge the imagines. Probably there were some difficulties in practice. However, the authors have made efforts to insert the GFP into the two domains to obtain a functional construct, if they want, it would be no doubt that they can accomplish such a colocalization assay.

Response: (Fig. 4g and Supplementary Fig. 4; Line 248-259) We agree with this reviewer's comment. In response, we have carried out a new set of experiments to examine the interaction between these two proteins using the BiFC assay. Our results (shown in Fig. 4g and Supplementary Fig. 4) show that these two proteins interact in vivo. In addition, we were able to confirm once more that the domain 2 of QKY protein is cytoplasmic, and not domain 1. We have described these results in the results section (line 248-259).

Comment 2: It is very interesting finding that QKY can affect differential distribution of SCM in H/N cells. In this work, authors provided different lines of evidence to demonstrate the role of QKY in SCM ubiquitination, degradation and distribution. However, it is still not clear how these interactions at protein level can alter the SCM differential distribution in epidermal cells. In the discussion, the authors mentioned the abundant regulation in protein distribution. In their data, there were some clues that the ratio of mono vs multi-ubiquitination or the rate of movement to vacuole degradation might contribute the abundant regulation of SCM distribution. However, additional to the role of the change in different genetic background

with genetic data, there are no direct evidences at cellular level to support the speculation.

Response: (Line 425-432) Our results show that QKY accumulates preferentially in the H-position epidermal cell and stabilizes SCM degradation by preventing its ubiquitination through their physical interaction. This initiates differential accumulation of SCM between the epidermal cells at two positions. Previously it was reported that SCM inhibits WER expression in the H-position cells (Kwak and Schiefelbein, 2007) and that WER negatively regulates SCM expression (Kwak and Schiefelbein, 2008), This feedback regulation seems to further establish the differential accumulation of SCM. However, it is not still clear how QKY accumulates preferentially in the H-position cell and how QKY prevents ubiquitination of SCM, which we believe are interesting future goals to be explored. We have explained this more clearly in the discussion section (line 425-432) of the revised manuscript.

Comment 3: The third issue concerned is about the ubiquitination sites. The authors demonstrated that at the interaction region between QKY and SCM, there are three ubiquitination sites. After all of three substituted, the function of SCM was lost. It is reasonably asking whether the all three amino acids required or only one or two of them are required? Any author hate to be asked to add more data. However, it is for sure that the authors are doing experiments to the questions as such question are too obvious to be addressed.

Response: (Supplementary Fig. 7; Line 343-354) In addition to our test of many mutated SCMs in the previous version of the manuscript, we have analyzed even more mutated SCMs for this revised manuscript, as requested by this reviewer. We found that the double mutated version K520/526R SCM-GFP showed similar mobility to that of non-ubiquitinated SCM-GFP (Supplementary Fig. 7). Unfortunately, we were not able to obtain the double mutated version K520/525R of this construct. Nevertheless, together with our previous results with K520R, K525R, K526R, K525/526R, and

K520/525/526R, our new results enable us to conclude that the Lys-520, together with at least one lysine residue at 525 and 526 are monoubiquitinated. These new findings and interpretations are described in the Results section of the revised manuscript (line 343-354).

Comment 4: Following the above concerns or comments, this manuscript read loosely organized. The part of mutant characterization is necessary for the work and well organized of the first three sessions of results. But the following parts seems diffused. Several issues were addressed, such as CPC movement, PD limitation and protein abundance regulation. None of these issues seems effectively investigated. If to focus the mechanistic investigation into the interaction between the QKY and SCM and propose one or two definitive conclusions on the role of QKY on SCM distribution, this work would be presented more coherently and readably

Response: In response to this suggestion, we have re-organized and shortened the discussion section to clarify the message.

Comment 5: The minor issue is the organization of writing of discussion part. It can be significantly shorten and concisely organized.

Response: As mentioned above, we have re-organized and shortened substantially the discussion section as the reviewer suggested.

<Reviewer 3>

Comment 1: First, although the authors focus on especially the root epidermal, the CPC movement assays are done in leaf cells. Why didn't the authors perform the movement assays in the physiological context? Is this because of a technical reason?

Response: We understand this reviewer's concern. Unfortunately, to investigate CPC movement in the root, it is necessary to introduce DNA into

the epidermal cells in the meristematic region. However, it is very difficult to use biolistics for genetic transformation in meristematic root tissues, because it is covered by lateral root cap cells. Fortunately, trichome development in the leaf epidermis is controlled by a similar genetic regulatory network to the network controlling the cell fates in the root epidermis (Ryu et al., 2013). Therefore, we examined the CPC movement between the epidermal cells in the leaf epidermis and used these results to infer the possible mechanism in the root epidermis.

Comment 2: My second major concern is most important analyses in this work lack quantification data, which are pointed out in the following comments:

Comment 2-1: Line 158. Figure 1b and the statement that QKY seems to be more abundant in H cells is not acceptable without some form of quantification. An optical section image is a better choice. Arrows should also be provided to guide the reader to different cells. Figure 1c, Figure 2a, 3a lack the quantification. I appreciate that it is in Table 1 but I don't think it is a great way to represent data. I think they should be in the same figure as a histogram, much better for visualisation.

Response: (Fig. 1b) We agree with this concern. Accordingly, we included cross-section images of the *QKYp:QKY-GFP* plant root and the *SCMp:SCM-GFP* plant root. We also reduced the propidium signal (red) to show the GFP signal more clearly. Now the images show the preferential localization of QKY-GFP in the *QKYp:QKY-GFP* plant root and the SCM-GFP in the *SCMp:SCM-GFP* plant root. better (Fig. 1b).

The suggested visualization with a histogram of the phenotype is a good idea, but we realized it would add too many duplications in figures. Therefore, we opted to present the quantified data as a table as a better way given the many quantified phenotype data.

Comment 2-2: The authors claimed “the CPC-GFP did not exhibit greater accumulation in the H position cells in *qky-16* and *scm-2*” in Line 202. It is

rather fundamental for their argument so strong evidence are needed for support. Images of Fig2b is not convincing for the conclusion without quantification.

Response: (Fig. 3b) As the reviewer suggested, we have now included statistical analyses on the CPC-GFP movement from the stele to the epidermis showing that preferential accumulation in the H-position cells is significantly impaired in the *qky-16* and the *scm-2* mutants (Fig. 3b).

Comment 3: Also, the data in leaves shown in Fig2c makes their arguments somewhat weaker as the authors was trying to explore the function of QKY in root, not in leaves. What kind of movement is then being suggested? GFP movement not affected, CPC-GFP affected. Are they speculating a facilitated movement with chaperones helping the passage of CPC? How does that relate to QKY ecc. How would they then reconcile this with the speculation that QKY involved in the import, rather than export of CPC? I find a gap and some confusion here in the manuscript about the actual mechanism of movement.

Response: (Line 201-214 and Line 376-380) As we explained in response to another reviewer comment (above), there are technical reasons for using leaf tissues to explore CPC movement instead root tissue. Second, we found that CPC-GFP movement is facilitated by QKY and SCM. The inability of co-expression of QKY or SCM to restore the CPC movement (Fig. 3c), and the preferential accumulation of QKY and SCM (Fig. 1b) led us to suggest their role in import rather than export. As the reviewer points out, a targeted movement or selective movement through PD requires protein-protein interaction. However, we were not able to detect the interaction of QKY and SCM with CPC using yeast two-hybrid assay (Supplementary Fig. 3). Therefore, it seems that QKY and SCM regulate CPC movement indirectly (through another factor(s)). We have mentioned indirect regulation in results section (Line 201-214) and in the discussion section (Line 376-380).

Comment 4: While analyzing SCM translation level, the authors using transgenic lines with *SCMp::SCM-GFP* in *qky-16scm-2* and *scm-2* background. They claimed the SCM promoter activity and SCM-GFP transcript level are similar in the two background (supp. Fig5). However, they did not show quantification of GUS and SCM-GFP transcript level in those transgenic lines, which is necessary for comparing SCM protein levels.

Response: (Supplementary Fig. 5a,c; Line 268-271) This is essentially the same comment as reviewer 1's comment 2.

We used the same transgenic line (*SCMp:SCM-GFP*) to make the *SCMp:SCM-GFP scm-2* plant and the *SCMp:SCM-GFP qky-16 scm-2* plant by genetic crosses, and we further showed that the promoter activity of SCM in the *qky-16* root was not significantly different from the activity in the wild-type root based on the *SCMp:GUS* activity introduced into those mutants by genetic crosses (Supplementary Fig. 5a). In addition, we also show that the steady state level of *SCM-GFP* transcript was not different between the *SCMp:SCM-GFP scm-2* mutant root and the *SCMp:SCM-GFP qky-16 scm-2* mutant root (Supplementary Fig. 5c). We have described this in the results section (line 268-271) of the revised manuscript. Together, these results enable us to exclude the possibility that the transgene behaves differently in these two lines.

Comment 5: Figure 4a the first image is clearly saturated. I would prefer a quantification for panel a,d, e, f (all the microscopy). I think that should be the standard these days for image analysis. That said I appreciate that having protein levels next to these confocal images is somewhat of a form of quantification. Similar argument for Fig 5c.

Response: (figure 2 below; Fig. 5d,f,g and Fig. 6c,d; Line 330) We agree that the first confocal image in figure 5a is clearly saturated. However, it is necessary to increase the detector gain quite high to show the SCM-GFP signal in the *qky-16 scm-2* mutant root (because of its weak signal) (see the

figure 2 below), and this setting leads to the oversaturation for the first image. However, this saturation doesn't seem to harm our main message because this result sufficiently shows the difference between these two lines. Furthermore, we also have the supporting western blot data (Fig. 5b). Also, we quantified the data which the reviewer pointed out (Fig. 5d,f,g and Fig. 6d) and we have now performed statistical analysis showing its significance. We have also carried out the western blot experiment using anti-GFP antibodies with total protein extracts from the same lines used for the confocal images as the reviewer suggested, which showed that the protein level from the *SCMp:SCM_{m22KR}-GFP qky-16 scm-2* plant root was similar to the protein level from the *SCMp:SCM_{m22KR}-GFP scm-2* plant root unlike the wild-type version of SCM-GFP (Fig. 6c).

Figure 2. GFP accumulation in the developing root meristem of various plants harboring *SCMp:SCM-GFP* translational fusions.

Comment 6: In western blotting assay, there were two protein bands reacting with anti-GFP antibody, which was explained by ubiquitination and reduced gel mobility. I would suggest the authors to confirm it is SCM-GFP using

another GFP antibody and SCM antibody.

Response: (figure 3 below) As the reviewer suggested, we performed the western blot analysis using different antibodies. Because anti-SCM antibodies are not available, we used different anti-GFP antibodies (Abcam, rabbit polyclonal to GFP, cat # ab290) from the one we used for the whole experiments (Clontech, mouse monoclonal to GFP, cat # 632375). As we expected, we obtained the same result as before with the monoclonal anti-GFP antibodies (see figure 3 below). Again, this western blot data shows that the level of SCM-GFP protein in the *SCMp:SCM-GFP qky-16 scm-2* plant root was much lower than in the *SCMp:SCM-GFP scm-2* plant root. Furthermore, the protein reacting with these antibodies in the *qky-16 scm-2* mutant root had higher molecular weight than the SCM-GFP protein in the *scm-2* mutant root as we observed before with the monoclonal anti-GFP antibodies.

Figure 3. Western blot analysis to examine the SCM-GFP level using different anti-GFP antibodies (Abcam, rabbit polyclonal to GFP, cat.# ab290).

Comment 7: All the data they have about the mechanism of internalisation are based on chemical treatments; I wonder if they could think of potential further evidence of a different nature?

Response: (Line 300-305) As the reviewer points out, we analyzed

internalization after treatment of TyrA23. To demonstrate clathrin-mediated endocytosis in a direct way, we need to use an inducible clathrin loss-of-function line (such as inducible dominant-negative HUB1), which is a more involved and long term experiment. However, we think that the TyrA23 treatment experiment is sufficient to show the internalization of SCM-GFP not only because the chemical treatment is generally accepted way to block endocytosis, but also because we showed the increased accumulation of SCM-GFP at the plasma membrane otherwise it accumulates and is degraded in the vacuole. To account for this reviewer's concern, we have changed the description 'clathrin-mediated endocytosis' to 'endocytosis' (line 300-305) .

Rererences

Geldner N, Denervaud-Tendon V, Hyman DL, Mayer U, Stierhof YD, Chory J. Rapid, combinatorial analysis of membrane compartments in intact plants with a multicolor marker set. *Plant J* 59, 169-178 (2009).

Kwak SH, Schiefelbein J. The role of the SCRAMBLED receptor-like kinase in patterning the Arabidopsis root epidermis. *Dev Biol* 302, 118-131 (2007).

Kwak SH, Schiefelbein J. A feedback mechanism controlling SCRAMBLED receptor accumulation and cell-type pattern in Arabidopsis. *Curr Biol* 18, 1949-1954 (2008).

Ryu KH, Zheng X, Huang L, Schiefelbein J. Computational modeling of epidermal cell fate determination systems. *Curr Opin Plant Biol* 16, 5-10 (2013)

Reviewers' comments:

Reviewer #1 (Remarks to the Author):

In this revised version, the authors have included additional data to answer most of the concerns raised by the reviewers. At this point, I still have a few points to address.

Follow-up to Comment 3

Fig 5e legend: "Co-localization of internalized SCM-GFP and mcherry-RabF2a. Seedlings were treated with E-64d (20 μ M) for 2 h. Arrows indicate co-localization of the two proteins. Bar = 10 μ m."

First of all, the explanation to the panels regarding the DMSO-treatment is missing in this legend.

Secondly, in the panels showing DMSO-treated samples, the structures marked by mCherry-RabF2a seems to be too large to be endosomes (compare the scale bar and the original publication of Geldner et al. 2009 Plant J). This could be due to the saturation of the signals in this particular image. In any case, this image is not representative for a late endosomal marker.

Thirdly, there is no colocalization between the GFP and mCherry signals in the DMSO-treated control and thus the arrows do not point to colocalizing endosomes in these panels. It is difficult to draw any conclusion without state-of-the-art quantification of the colocalization efficiency. However, there should be an explanation in the text as to why SCM-GFP does not colocalize with the late endosome marker in the absence of E64-d. If SCM-GFP is transported to the vacuole by endosomes, it would be expected that there is at least a partial overlap between the GFP- and mCherry-marked structures even without E64-d treatment.

Follow-up to Comment 4

Lines 300-305: The authors might want to consider citing the Dejonghe et al. 2016 Nature Comm. Paper to explain the rationale of using TyrA23.

Follow-up to Comment 5

Lines 420-422: "Therefore, this study provides evidence that multi-monoubiquitination can serve as both an internalization signal and a vacuolar targeting signal for degradation of plant proteins."
"

In the text (lines 415-422), the authors rephrased the paragraph, which reads now much better. I would therefore suggest to remove this last sentence of the paragraph, since this statement is not unambiguously supported by the experimental data that the authors show. As the authors write in the revised text, the fact that they could not detect clear polyubiquitination is not an evidence that multi-monoubiquitination is indeed the signal for targeting/degradation. The authors have shown by the lysine-mutations that ubiquitination is required, but the identity of the type of ubiquitination remains to be determined.

Reviewer #2 (Remarks to the Author):

The revision properly responded my concerns. I believe that this manuscript could be accepted as is

Reviewer #3 (Remarks to the Author):

The authors provided more strong data according to our comments in the revised manuscript. I don't have more comments.

Response to the Reviewer #1

In response to the additional suggestions made by Reviewer #1, we were able to increase the quality of our analysis of the co-localization of two proteins (SCM-GFP and mCherry-RabF2a) through an approach suggested by the reviewer. We have modified our manuscript accordingly. This has led to the separation of the original Figure 5 into two figures (Figure 5 and Figure 6) in this revised manuscript, due to the additional information exceeding the word limit for one figure legend.

Our detailed response follows.

Comment 1: Fig 5e legend: “Co-localization of internalized SCM-GFP and mcherry-RabF2a. Seedlings were treated with E-64d (20 μ M) for 2 h. Arrows indicate co-localization of the two proteins. Bar = 10 μ m.”

First of all, the explanation to the panels regarding the DMSO-treatment is missing in this legend.

Response: (Fig. 6b; Line 854-855) We have modified the legend and this figure.

In our original figure, we made a mistake with the placement of the arrows. Some of the arrows should not have been placed in the control sample (DMSO). We decided to remove all arrows in figure 6b because too many arrows interfere with the fluorescence signals, especially in the E-64d treated sample. As a result, we rephrased the figure legend to make it clear that the DMSO treated sample is the samples without E-64d. ‘Seedlings were treated with or without E-64d (20 μ M) for 2 h.’

Comment 2: “Secondly, in the panels showing DMSO-treated samples, the structures marked by mCherry-RabF2a seems to be too large to be endosomes (compare the scale bar and the original publication of Geldner et al. 2009 Plant J). This could be due to the saturation of the signals in this particular image. In any case, this image is not representative for a late endosomal marker.”

Response: (Fig. 6b; Line 289-297; Line 531-535) We appreciate this suggestion. In our original images, we simply tried to show the signals as strong as possible. Since we used large pinhole in the confocal microscopy to get these strong signals, the fluorescence images were somewhat blurry and gave large spots which made it difficult to assess co-localization. As the reviewer suggested, we took new fluorescence images using a recommended pinhole size and were able to quantitatively analyze the co-localization using Image-pro 10 software. As the reviewer predicted, we were able to detect a small portion of the internal SCM-GFP co-localized with mCherry-RabF2a in the control which was not treated with E-64d (DMSO). In the E-64d treated seedlings, the internal SCM-GFP strongly co-localized with mCherry-RabF2a. We obtained the Pearson correlation coefficient (r) for the association between these two proteins by analyzing the fluorescence signals inside 5 different cells in one seedling. We showed the average value \pm SD of 5 independent Pearson correlation coefficients from 5 different seedlings in a graph.

Comment 3: “Thirdly, there is no colocalization between the GFP and mCherry signals in the DMSO-treated control and thus the arrows do not point to colocalizing endosomes in these panels. It is difficult to draw any conclusion without state-of-the-art quantification of the colocalization efficiency. However, there should be an explanation in the text as to why SCM-GFP does not colocalize with the late endosome marker in the absence of E64-d. If SCM-GFP is transported to the vacuole by endosomes, it would be expected that there is at least a partial overlap between the GFP- and mCherry-marked structures even without E64-d treatment.”

Response: (Fig. 6b; Line 289-297; Line 531-535) As mentioned above, we are now able to clearly show a small portion of the SCM-GFP co-localizes with mCherry-RabF2a.

Comment 4: Follow-up to Comment 4: Lines 300-305: “The authors might

want to consider citing the Dejonghe et al. 2016 Nature Comm. Paper to explain the rational of using TyrA23.”

Response: (Line 302, 678) We have included this paper (Dejonghe et al., 2016) in our References.

Comment 5: Follow-up to Comment 5: Lines 420-422: “Therefore, this study provides evidence that multi-monoubiquitination can serve as both an internalization signal and a vacuolar targeting signal for degradation of plant proteins. “

In the text (lines 415-422), the authors rephrased the paragraph, which reads now much better. I would therefore suggest to remove this last sentence of the paragraph, since this statement is not unambiguously supported by the experimental data that the authors show. As the authors write in the revised text, the fact that they could not detect clear polyubiquitination is not an evidence that multi-monoubiquitination is indeed the signal for targeting/degradation. The authors have shown by the lysine-mutations that ubiquitination is required, but the identity of the type of ubiquitination remains to be determined.

Response: (Line 422) As the reviewer suggested, we have removed the last sentence of this paragraph.

REVIEWERS' COMMENTS:

Reviewer #1 (Remarks to the Author):

In this revised manuscript, the authors provided new data on the colocalisation of SCM-GFP and the endosome marker and answered all the points raised by this reviewer.

The manuscript convincingly shows the functional relationships between QRY, SCM and CPC and provides novel insights into the regulation of root epidermal patterning.